# RE-IMAGINE: Symbolic Benchmark Synthesis for Reasoning Evaluation

Xinnuo Xu [* 1]   Rachel Lawrence [* 1]   Kshitij Dubey [* 2]   Atharva Pandey [* 2]   Risa Ueno [1]   Fabian Falck [1]
Aditya V. Nori [1]   Rahul Sharma [2]   Amit Sharma [2]   Javier Gonzalez [1]

## Abstract

Recent Large Language Models (LLMs) have reported high accuracy on reasoning benchmarks. However, it is still unclear whether the observed results arise from true "reasoning" or from statistical recall of the training set. Inspired by the ladder of causation (Pearl, 2009) and its three levels (associations, interventions and counterfactuals), this paper introduces RE-IMAGINE: a framework to characterize a hierarchy of reasoning ability in LLMs, alongside a scalable pipeline to generate problem variations across all the levels of the hierarchy. By altering problems in an intermediate symbolic representation, RE-IMAGINE generates arbitrarily many problems that are not solvable using memorization alone. The framework is general and can work across reasoning domains, including math, code, and logic. We demonstrate the type of insights that RE-IMAGINE can generate on four widely-used benchmarks, which we use to evaluate reasoning on several families of LLMs. We observe reductions in performance when the models are queried with problem variations. These assessments indicate a degree of reliance on statistical recall for past performance, and open the door to further research targeting skills across the reasoning hierarchy.

## 1. Introduction

Recent advancements in Artificial Intelligence (AI) have sparked increasing interest in the development of reasoning systems. Central to this goal are Large Language Models (LLMs) – models like OpenAI's o1 (Jaech et al., 2024), o3 (OpenAI, 2024), or DeepSeek-R1 (Team, 2025) show complex problem-solving abilities, and demonstrate unprecedented results on reasoning benchmarks, e.g. FrontierMath (Glazer et al., 2024) and ARC-AGI (Chollet et al., 2024). With growing reliance on LLMs across wide-ranging applications, it is increasingly important to clarify the strengths and limitations of these apparent reasoning abilities.

Reasoning is a cognitive process. It involves using facts or premises to make inferences about conclusions or judgments (Holyoak & Morrison, 2005). In the realm of LLMs and AI, reasoning is understood to be the ability of a model to demonstrate logically correct systematic processes that surpass mere statistical pattern recognition in the training set (González & Nori, 2024).

Traditionally, the evaluation of reasoning in LLMs has been focused on their performance across ***fixed*** benchmarks in domains such as math (Cobbe et al., 2021), programming (Wu et al., 2024; Gu et al., 2024), real-world logic (Jin et al., 2023a) and others (Wang et al., 2019; Hendrycks et al., 2020; Srivastava et al., 2022). However, debate persists on whether the observed results occur from genuine reasoning or from mere statistical recall of training data (Mitchell & Krakauer, 2023) – particularly for training data which, in the case of published benchmarks, may have information leakage from the test set (Zhou et al., 2023). Defining principled ways to make this distinction is crucial for advancing AI and controlling potential hazards and risks (Weidinger et al., 2021).

Recently, several surveys have explored how to evaluate reasoning beyond memorization in LLMs (Xu et al., 2025; Huang & Chang, 2023). In general, two main approaches have emerged. One aims to develop novel reasoning tasks such as mystery blocksworld (Webb et al., 2024), ARC-AGI (Chollet et al., 2025), and others (Zhu et al., 2023). An alternative is to create novel variations of existing benchmarks, e.g. for math (Mirzadeh et al., 2024; Srivastava et al., 2024), analogies (Lewis & Mitchell, 2024), and diverse tasks across code, math, and logic (Wu et al., 2023; Zhang et al., 2024). A common strategy for creating such variations involves leveraging symbolic representations of problems such as functional templates (Mirzadeh et al., 2024; Srivastava et al., 2024), reasoning or causal graphs (González & Nori, 2024; Hüyük et al., 2024; Yang et al., 2024), planning tasks (Valmeekam et al., 2022) or code (Li et al., 2024).

---
*Equal contribution [1]Microsoft Research Cambridge, UK [2]Microsoft Research India, India. Correspondence to: Xinnuo Xu <xinnuoxu@microsoft.com>, Rachel Lawrence <rachel.lawrence@microsoft.com>.

*Proceedings of the 42nd International Conference on Machine Learning*, Vancouver, Canada. PMLR 267, 2025. Copyright 2025 by the author(s).

| Level | Description | Examples | Evaluation metric(s) | References |
|---|---|---|---|---|
| **1. Observe** | Original problem. | Problems in GSM8K, CLadder, CRUXEval Loop | Task performance (original). | Cobbe et al. (2021) Jin et al. (2023a) Kamath et al. (2024) Gu et al. (2024) |
| **2. Mutate** | Mutated problem by replacing or adding components. | Replacing numeric value, changing variable name, modifying operator, irrelevant information | Task performance (original and mutated). | Mirzadeh et al. (2024) Srivastava et al. (2024) Wu et al. (2023) Lewis & Mitchell (2024) |
| **3. Imagine** | Original problem augmented with an *'imagine'* statement, modifying the original statements or assertions before it. | Extra logic involving revisions or counterfactual statements. | Task performance (original and augmented) *PN* and *PS* (only in counterfactuals). | González & Nori (2024) Hüyük et al. (2024) |

*Table 1.* Hierarchy of problem variations introduced in the RE-IMAGINE framework to evaluate LLMs. *Level-1 (observe)* captures the accuracy of LLMs to solve problems in existing benchmarks. These benchmarks may have been *observed* by the LLMs in current or similar form. *Level-2 (mutate)* captures the ability to solve problem variations. *Level-3 (imagine)* captures the ability to correctly incorporate new logic into existing problems, even when this logic contradicts the original components. In certain cases, the new logic can be understood as a counterfactual statement (Pearl, 2009).

Despite the introduction of various benchmark variations aimed at assessing LLMs' reasoning abilities, these variations have been developed in an ad hoc manner, lacking a systematic hierarchy. Moreover, most existing approaches rely on significant manual effort and are designed for specific tasks, making them difficult to scale across multiple benchmarks and tasks. For instance, in Mirzadeh et al. (2024); Srivastava et al. (2024), functional templates for simple math problems in the GSM8K benchmark (Cobbe et al., 2021) are manually created, restricting the analysis to only 100 new templates.

Inspired by Judea Pearl and his ladder of causation (Pearl, 2009), our work expands on the traditional way to evaluate reasoning. Pearl asserts that: *"Only machines that can correctly perform correlations, interventions and counterfactuals will have reasoning abilities comparable to humans."* Following this principle, we present a new framework, RE-IMAGINE, to characterize a hierarchy of reasoning abilities in LLMs, alongside an automated pipeline to guarantee scalable evaluations in each level of the hierarchy.

RE-IMAGINE generalizes, expands and scales up evaluation of LLM reasoning by means of an pipeline with three core components:

(i) A language-to-code model that converts each benchmark question into a symbolic (code) representation.
(ii) A set of mutations of the symbolic representation that creates an 'executable' variation of the problem.
(iii) A code-to-language component that translates the generated symbolic problems back into natural language.

The intermediate executable symbolic representation ensures that correct outcomes can be calculated automatically from the mutations. This approach generates a diverse set of "unseen" variations of existing, well-established benchmarks, providing novel challenges for LLMs.

RE-IMAGINE enables us to *reinvent* the standard approach to evaluating reasoning in LLMs. As our experiments show, benchmarks across domains such as math, code, and logic can be systematically transformed using the same principles, generating challenging new scenarios that are unlikely to appear in the LLMs' pre-training data. Our findings indicate that all tested LLMs exhibit some degree of reliance on statistical recall, while problems at higher levels in the reasoning hierarchy remain a yet-unsolved challenge.

**Contributions.** This paper presents three major contributions, corresponding to each of the following sections:

In **Section 2**, we propose a hierarchical framework to characterize existing and new approaches for evaluating reasoning in LLMs. The proposed hierarchy has three levels of increasingly difficulty that capture different levels of reasoning via variations of the problems of existing, well-established benchmarks.

In **Section 3**, we propose an end-to-end, *scalable* pipeline that allows the generation of an arbitrary number of new problems in each level of the hierarchy. This is crucial to scale up current approaches that require the manual generation of new scenarios.

In **Sections 4-5**, we use RE-IMAGINE to re-analyze the reasoning abilities of all models in the GPT (Brown et al., 2020), Llama (Touvron et al., 2023), and Phi families (Kambhampati et al., 2024). We focus on four reasoning benchmarks: GSM8K for math (Cobbe et al., 2021), CLadder for causality (Jin et al., 2023a), and CRUXEval (Gu et al., 2024) and Loop (Kamath et al., 2024) for code. We show consistent decline in LLM performance as reasoning complexity increases across all evaluated benchmarks.

**Novelty.** This paper introduces two major innovations:

The unified 3-level reasoning hierarchy presented in this paper incorporates both previously studied mutations and the newly introduced ones. According to this hierarchy, we highlight that prior research has mainly focused on Level-2 mutations, which assess a model's ability to generalize be-

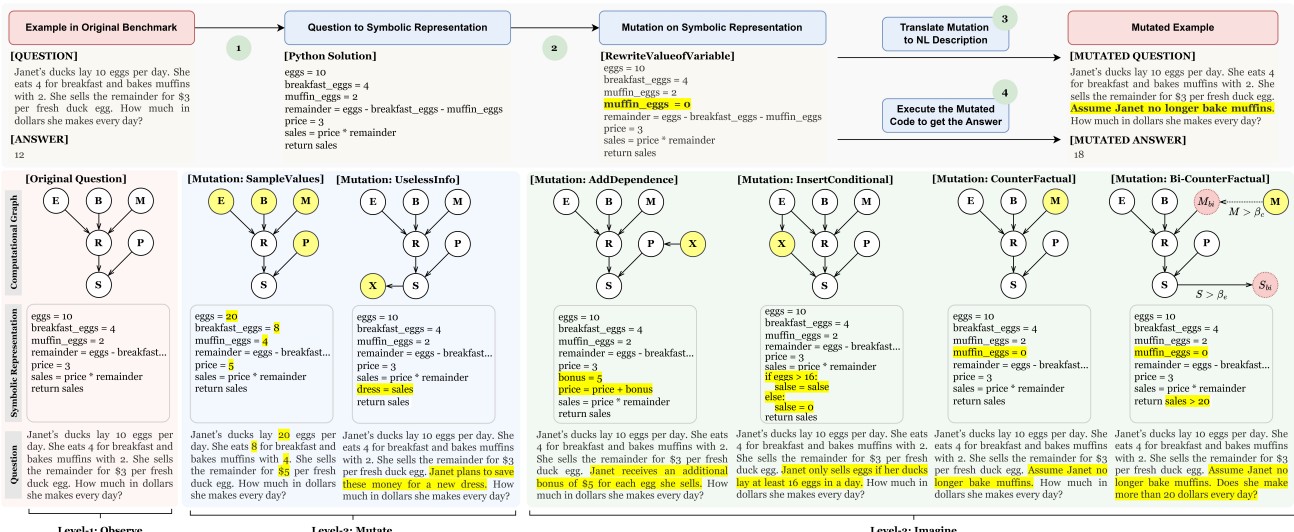

*Figure 1.* **Top**: The benchmark transformation pipeline (Section 3) outlined with an example from GSM8K (Cobbe et al., 2021). This pipeline leverages the symbolic representation of the question (a Python snippet form) to automatically transform a math QA problem (*leftmost*) into a similar format with additional reasoning steps (*rightmost*). **Bottom**: To clearly define the mutations, we transform the symbolic representation into a computational graph (*leftmost*). Nodes represent variables from the symbolic representation, and edges illustrate the dependencies between them. The remaining six graphs depict the different types of mutations (see Section 3.2). *E*, *B*, *M*, *R*, *P*, and *S* represent variable *eggs*, *breakfast_eggs*, *muffin_eggs*, *reminder*, *price*, and *sales*, respectively. The yellow highlights indicate the modifications made relative to the original versions. Red nodes represent binary values, while all other nodes are numerical.

yond existing benchmarks while maintaining the original reasoning path of the questions. In contrast, we emphasize that the Level-3 mutations introduced in our work are significantly more challenging.

Alongside the reasoning hierarchy, we introduce – to the best of our knowledge – the first scalable mutation generation pipeline that applies across multiple benchmarks and tasks. This framework enables the creation of an arbitrary number of mutations at each level of the hierarchy for existing benchmark problems.

A detailed comparison with previous studies can be found in Table 4 in Appendix B.

## 2. RE-IMAGINE: The *Ladder* of Reasoning

Inspired by the ladder of causation (Pearl, 2009), we systematically define a three-layer hierarchy ('observe', 'mutate', 'imagine') that characterizes different levels of reasoning abilities in LLMs, in the same way that the ladder of causation captures three different cognition skills. This allows us to characterize and compare the goals of different evaluation experiments with precision, both new and existing. A summary of the three levels is presented in Table 1.[1]

- **Level-1 ("Observe")** captures the accuracy (or other metric of interest) of LLMs on existing benchmarks. It is called *observe* because it is expected that an LLM which

has already *seen* training-set problems similar to the ones in the benchmark should be able to produce high accuracy.

- **Level-2 ("Mutate")** captures the ability of LLMs to solve problems that have been *mutated* by, for example, adding irrelevant information, renaming values, or changing values. It tests the ability of models to generalize beyond the existing benchmarks in cases where the core logical requirements of the questions are preserved. Several works have proposed approaches that sit in this level (Mirzadeh et al., 2024; Srivastava et al., 2024; Wu et al., 2023; Lewis & Mitchell, 2024), which are based on manually created functional variations of the original problems. The results in such variations highlight memorization and over-fitting issues. For a true reasoning model, the task performance should be invariant w.r.t. the class of changes in this level.

- **Level-3 ("Imagine")** is the topmost and most sophisticated level. It captures the models' ability to correctly incorporate new information and logic into existing problems. Given a problem defined by a set of logical predicates or facts, this variation *augments* the original problem with an additional predicate that changes some previously stated one. Correctly incorporating new logic requires an accurate (explicit or implicit) representation of the steps required to solve the problem, as well the ability to contradict and revise prior knowledge. Counterfactual assessments (González & Nori, 2024) sit on this level of the hierarchy. Task performance metrics and counterfactual related metrics like the probability of necessity (*PN*) and sufficiency (*PS*) can be used in this level as in (González

---

[1]We provide a more detailed explanation of the fundamental connection between our hierarchy and causality in Appendix C.

& Nori, 2024).

Next, we detail how to use the problems from benchmarks in **Level-1** to create novel problems in **Level-2** and **Level-3**.

## 3. Benchmark Synthesis Pipeline

We present a unified benchmark synthesis pipeline that automatically generates variations of existing benchmarks, preserving the core logic of the task while demanding stronger reasoning abilities to solve.

We illustrate the pipeline with a real example from the GSM8K benchmark, presented in the top row of Figure 1. Starting with a sample from the existing dataset, we first convert the question into **executable** symbolic form, e.g. code, knowledge graph (① in Figure 1). In line with Toshniwal et al. (2024), we represent the math question as a Python code snippet. Next, we apply a specific type of mutation to the symbolic representation (②). In this example, we overwrite the value of the variable *muffin_eggs* in the code. To maintain the core logic of the task – a natural language-based math question-answering (QA) problem – we translate the change in the symbolic representation back into natural language (NL) and incorporate it into the original question (③). Due to the executable nature of the Python code, the ground-truth answer for the mutated question can be obtained by running the modified code snippet (④).

Note that although Figure 1 illustrates all the elements in the pipeline, not all of them will be necessary for every benchmark of interest. A summary of the required steps for the benchmarks discussed in this paper is provided in Table 2. Similarly, not all mutations are applicable to every problem, and additional mutations beyond those listed in Figure 1 can also be considered.

### 3.1. NL-to-Symbolic (①) & Symbolic-to-NL (③)

Steps ① and ③, corresponding to transformations from NL to Symbolic and Symbolic to NL respectively, are non-trivial. Benchmarks that have original problems already in code form (e.g. Loop and CRUXEval) do not require these steps. When they are required, they need some level of adaptation to the nature of the benchmark. Details for these two steps are provided in Section 4.

### 3.2. Symbolic Mutations (②)

We showcase six code mutations spanning the **Level-2** and **Level-3** reasoning levels to create benchmark variations. To thoroughly define the mutations in ②, we first convert the symbolic representation into a computational graph (leftmost column in the bottom row of Figure 1). Nodes represent variables in the symbolic representation and edges capture their dependencies. The mutation applied to the computational graph is then reflected in the symbolic repre-

sentation and translated into NL. The remaining six columns in Figure 1 illustrate mutation variations.

### Level-2 mutations

- *SampleValues* assigns new values to all root nodes. When translating the mutation back to NL, only the values in the question are replaced with the new ones, while the rest of the narration remains unchanged. This mutation specifically aims to differentiate the model's reasoning ability from memorization caused by data contamination.

- *UselessInfo* adds a new node dependent on a randomly selected node from the original graph, with the change described in NL between the context and the question. This introduces additional context, but does not alter original statements or impact the correct answer. This assesses the model's ability to disregard irrelevant information.

### Level-3 mutations

- ***AddDependence*** also introduces a new node into the graph. However, unlike *UselessInfo*, a randomly selected node from the original graph is modified to depend on the new node for its calculation. This is likely to influence the correct answer to the question. A natural way to encode this mutation in NL is to append a statement to the end of the original question, amending the original statement context, making this a **Level-3** mutation.

- ***InsertConditional*** adds a new node that connects two non-adjacent nodes in the graph, with edges linking the first node to the new node and the new node to the second node. In symbolic terms, this mutation is represented as an if-else condition. Two variables are randomly chosen, and one variable's value is set to 0 depending on the value of the other. Describing this in NL as a change to the previous method of calculating the variable, it also becomes a **Level-3** mutation.

- ***CounterFactual*** randomly selects a node in the graph and overwrites its value. Unlike *SampleValues*, this mutation does not directly replace the number in the question. Instead, it presents the change as an assumption statement appended to the original question. Thus, it modifies an existing statement in the context and adds an extra reasoning step to the original question, making it a **Level-3** mutation.

- ***Bi-CounterFactual*** builds on *CounterFactual* to evaluate the model's ability to connect the presence or absence of a cause with its effect, an essential reasoning skill from the perspective of causation (Neuberg, 2003; Halpern & Pearl, 2005). Previous work (González & Nori, 2024; Hüyük et al., 2024) quantitatively evaluates this through *necessity* and *sufficiency inconsistency rates* (*N-IR* and *S-IR*), but relies on manually crafted questions and their counterfactuals. In contrast, our scalable pipeline unlocks

| Benchmark | Type | Input | Output | Required Steps |
|-----------|------|-------|--------|----------------|
| GSM8K | Mathematics QA | NL math question | Numerical answer | ① ② ③ ④ |
| CLadder | Causal QA | NL causal query | Y/N answer | ① ② ③ ④ |
| CRUXEval | Code understanding | Python function and input | Execution output | ② ④ |
| Loop | Loop invariant inference | C Code and an assertion | Y/N answer | ② ④ |

*Table 2.* A summary of the steps used in GSM8K (Cobbe et al., 2021), CLadder (Jin et al., 2023a), CRUXEval (Gu et al., 2024), and Loop (Kamath et al., 2024). Since the inputs for both CRUXEval and Loop are code snippets, steps ① and ③ are not required.

large-scale analysis. In *Bi-CounterFactual*, the computational graph is treated as a Structural Causal Model (SCM), where the overwritten node acts as the cause and the final answer (leaf node) serves as the effect. Specifically, *Bi-CounterFactual* requires binary cause and effect nodes, with the overwritten value ensuring a change in the cause statement's presence or absence.

In the next two sections, we apply our automatic benchmark synthesis pipeline to the math reasoning benchmarks GSM8K and CLadder (Section 4) and the code understanding benchmarks CRUXEval and Loop (Section 5).

## 4. Math Benchmarks: GSM8K and CLadder

This section first details the benchmark transformation process for GSM8K and evaluates its quality (Section 4.1). We then analyze model accuracy on numerical math questions using the first five mutations outlined in Section 3.2, excluding *Bi-CounterFactual* (Section 4.2). Since *Bi-CounterFactual* involves binary questions and is primarily assessed with causation metrics, it is discussed separately in Section 4.3. Section 4.4 demonstrates that the findings from GSM8K extend to CLadder, another math reasoning benchmark focused on probabilities and causality.

### 4.1. Transformation Pipeline

**Question to Symbolic Representation ①**   Toshniwal et al. (2024) introduced OpenMathInstruct, whose validation set contains 970 GSM8K QA examples paired with Python solutions generated by Mixtral-8x7B (Jiang et al., 2024). We construct our test set by filtering out examples where the Python solution execution does not match the ground-truth answers. To ensure high-quality mutations, we further filter the data to keep only those where all constant variables in the code (root nodes in the computational graph) align with the numbers in the question and vice versa.[2]

**Symbolic Representation to Mutation ②**   We incorporate all six types of mutation described in Section 3.2. Since most GSM8K questions are framed within a story context, we ensure that newly sampled values for existing variables align with the original value's type (float/integer) and sign to preserve the story's coherence. Within this constraint,

integers are sampled from a discrete uniform distribution, while floats are drawn from a uniform distribution centered around the original value. We also ensure that the final answer maintains the same type and sign as the original ground-truth answer.

**Mutated Symbolic to NL ③**   In the *SampleValues* mutation, only the values in the questions are replaced with newly sampled ones, while the rest of the narrative stays the same, so no new NL descriptions are required. For the other mutation types, we provide the original math question, its Python solution, and the code modifications to a LLM (GPT-4o), leveraging its text generation capabilities to describe the code changes in natural language. To guarantee the symbolic-to-NL translation is correct, we prompt GPT-4o a second time to back-translate the mutated math problem into Python by modifying the original question's Python solution. The generated code must produce an execution result that matches the ground truth answer of the mutated question. (Detailed prompts are shown in Figure 17 and Figure 16 in Appendix E.)

To verify the accuracy of the mutated QA pairs, we manually reviewed 50 randomly selected examples from each mutation type. Valid examples are the ones that contain a clearly defined question and a correct ground-truth answer. The percentage of invalid QA pairs in *SampleValues*, *UselessInfo*, *CounterFactual*, *AddDependence*, and *InsertConditional* were 3.33%, 0.00%, 6.67%, 5.00%, and 5.00%, respectively.

### 4.2. Reasoning on Numerical Math QA

In line with previous studies, during testing, all models are provided with 8 in-context examples with Chain-of-Thought (CoT) to help them understand the task (prompt shown in Figure 18 in Appendix E).[3] We evaluate models from three popular families—*Phi*, *Llama*, and *GPT* (see Table 3 in Appendix A for details).

The answer accuracies are presented in Figure 2, with the percentage of invalid mutated examples in each mutation category displayed as a hashed block above each bar. By assuming that models would correctly answer these questions if the QA pairs were valid, this estimation serves as an

---

[2]We account for commonsense numerical facts, such as one year having 12 months, and one hour containing 60 minutes.

[3]We follow this blog post to strengthen our prompt.

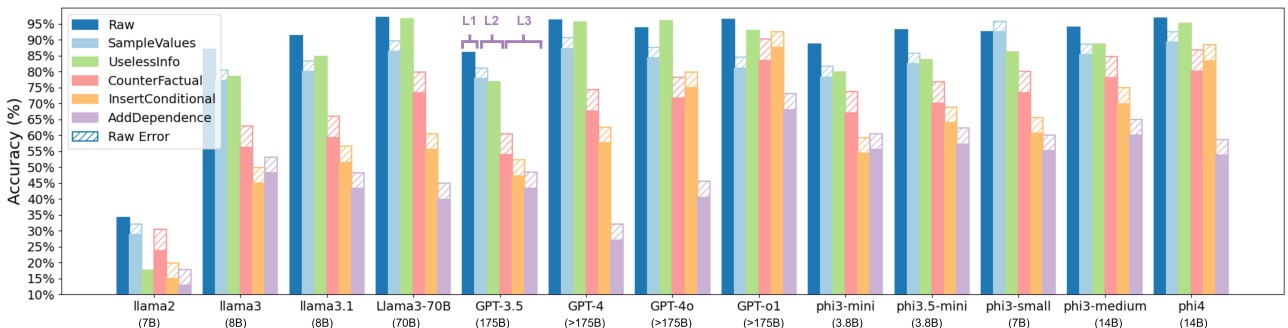

*Figure 2.* GSM8K results summary: model accuracy on numerical answer predictions across test set variations in different reasoning levels (see Section 3.2 and Figure 1). Due to potential noise from the automatic mutation process, we performed a human evaluation on the mutated test set. We added the percentage of invalid examples in each mutation category to the top of each accuracy bar as hashed blocks, assuming that if these QA pairs were correct, the models would answer them correctly. This provides an upper bound on the model's performance. We also present a box plot (see Figure 10 in Appendix E) to illustrate the statistical accuracy across 10 sets of samples.

upper bound on the model performance.

Key findings are: (1) With 8-shot in-context examples, most of the models achieve high accuracy ($\sim$ 95%) on the Level-1 raw test set. (2) Among **Level-2** mutations, *UselessInfo* is less challenging—especially for larger models—indicating their ability to ignore irrelevant details. However, nearly all models experience around 10% accuracy drop on *Sample-Values*, despite unchanged reasoning paths and only altered values. (3) **Level-3** mutations pose a greater challenge, with models showing significantly lower upper-bound performance than on **Level-1** and **2** test sets.

We also conduct ablation experiments to examine performance on test set containing multiple mutations (Appendix E.3). We found that composing mutations increases performance gap between the mutated and the original sets.

### 4.3. Reasoning Evaluation with Binary Counterfactuals

*Bi-CounterFactual* as described in Section 3.2 creates two auxiliary nodes in the computation graph with binary versions of a condition and an outcome. The reason for considering this problem transformation is that it allow us to compute metrics that are relevant to evaluate reasoning beyond accuracy. As shown in González & Nori (2024); Hüyük et al. (2024), this scenario enables the computation of the probabilities of necessity (*PN*) and sufficiency (*PS*) from the counterfactual literature (Pearl et al., 2000). Intuitively, these measures capture the probability of activating/deactivating a binary outcome in the presence/absence of a binary input. Although this restricts our analysis to the simplified (binarized) version of the GSM8k introduced by the *Binary Counterfactual* mutations, we compute these metrics due to their intrinsic value.

The ground truth *PN* and *PS* varies across problems and nodes. To give a benchmark-level measure of how well different models approximate them, we use the necessity

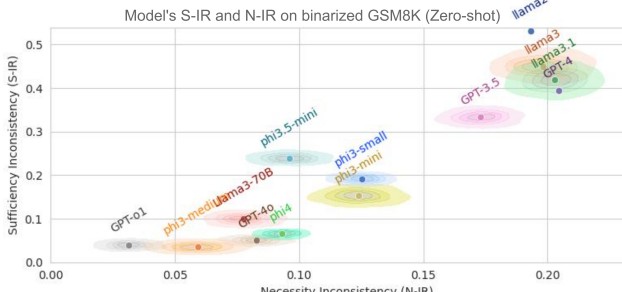

*Figure 3.* *Sufficiency/necessity inconsistency rates (S-IR/N-IR) on GSM8K factual/counterfactual test set. Models located near the bottom-left corner are thought to predict the causal relationship between the cause and effect, i.e. sufficient and necessary, in a way that is consistent with the true causal relationship, as defined by the ground truth.*

and sufficiency inconsistency rates (*N-IR*, *S-IR*) introduced in Hüyük et al. (2024), which account for the errors in the approximation of these measures in a normalized way (an optimal reasoning LLMs is one with *N-IR* = 0 and *S-IR* = 0). Because obtaining *N-IR* and *S-IR* is computationally expensive, we used 50 questions from the validation set of the benchmark, where the condition node was randomly sampled across the available leaf nodes. To obtain these results, we follow the same setup as in Hüyük et al. (2024).

Figure 3 shows the average *S-IR* and *N-IR* for all models across 50 random examples. Consistent with the numerical accuracy evaluation, GPT-o1 remains the best-performing model, while Llama 8B models, GPT-3.5, and GPT-4 are at the other end of the spectrum. However, GPT-4o and Llama 70B outperform phi3-small and phi3-mini in the causal reasoning evaluation.

### 4.4. Similar Findings on CLadder

To confirm that our findings in GSM8K generalize to other math reasoning benchmarks, we use the transformation pipeline (Section 3) to automatically generate three test set variations for CLadder, a causal reasoning benchmark.

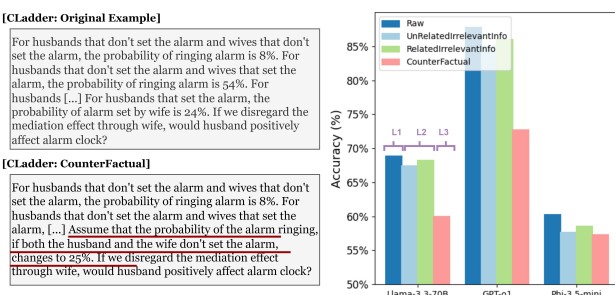

Figure 4. *Left*: An example from CLadder test set and its *Counter-Factual* mutation. **Right**: CLadder accuracy of the best-performing model in each family on causal question answering across test set variations at different reasoning levels. The full results can be found in Figure 21 in Appendix F. *UnRelatedIrrelevantInfo* and *RelatedIrrelevantInfo* are categorized as *UselessInfo* per Figure 1.

We examine three mutations *UnRelatedIrrelevantInfo*, *RelatedIrrelevantInfo*, and *CounterFactual*, with the first two classified under *UselessInfo* (see Figure 1 and Section 3.2). Details on the benchmark, implementation, LLM prompts, and mutation specifics are provided in Appendix F. The accuracy of the best-performing model in each family is shown in Figure 4. Similar to our observations in GSM8K, *Level-3* mutations present a greater challenge than *Level-2*, causing models to exhibit an approximately 20% drop in accuracy relative to the original test set. Models exhibit lower accuracy on test set variations composed of two types of mutations (Figure 21 in Appendix F).

# 5. Code Benchmarks: CRUXEval and Loop

We investigate two code understanding tasks: input/output prediction (CRUXEval) and automatic inference of loop invariants (Loop).

The CRUXEval benchmark (Gu et al., 2024) consists of 800 short LLM-generated Python functions alongside an input-output pair for each function (see Figure 5 for an example). The model's task is to predict the output of the function when evaluated on a given input parameter. Functions are filtered to include only those with low computation and memory requirements, with no side-effects or randomization, and excluded arithmetic calculation.

The Loop benchmark (Kamath et al., 2024) contains loop invariant inference tasks, each of which consists of a program with a loop and an assertion (see Figure 6 for an example). The goal is to infer a predicate that satisfies the following three conditions: it holds before the loop starts executing, holds for each iteration of the loop, and implies the assertion when the loop exits.[4] The model succeeds

---

[4] Although inferring such loop invariants is undecidable, in practice, checking whether a given candidate invariant satisfies the three conditions can be done well by automated software verification tools like Frama-C (Correnson et al.).

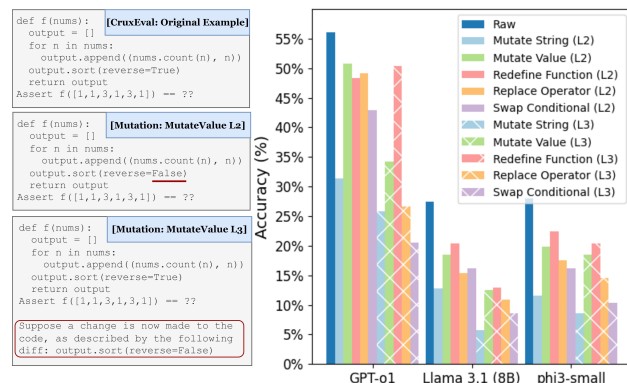

Figure 5. *Left*: An example from CRUXEval dataset and the *Level-2* and *Level-3* versions of its *MutateValue* mutation. **Right**: CRUX-Eval accuracy of the best-performing model in each family. Full results can be found in Figure 23 of Appendix G.

on the task if Frama-C (Correnson et al.) verifies via SMT solvers (de Moura & Bjørner, 2008) that the LLM output is a loop invariant, and fails otherwise (details in Appendix H).

Since both tasks are already programs, steps ① and ③ of the pipeline are not required.

## 5.1. Symbolic Mutations to CRUXEval ②

Each CRUXEval mutation can be applied either directly to the function code (*Level-2*), or presented as a code diff in an "imagine" statement following the original code (*Level-3*). The *Level-2* versions, in particular, are designed to have minimal effect on code length, execution time, and overall question difficulty. Table 6 in Appendix G shows the different types of mutations implemented for CRUXEval, summarized below.

- **Replace Operator** replaces an operator of type `ast.BinOp`, `ast.UnaryOp`, `ast.BoolOp`, or `ast.AugAssign` with its inverse or negation, where applicable.

- **Mutate String** replaces a string instance with a uniformly random character sequence of the same length.

- **Mutate Value** changes the value of an instance of type `bool`, `int`, or `float`. Booleans are replaced with their negations; integers / floats are perturbed by a uniformly random nonzero integer / float (respectively) in $[-10, 10]$.

- **Swap Conditional** selects a random conditional node for modification. If both an `if` and `else` branch are present, the code body for each branch is swapped. If only an `if` branch is present, the condition of the branch is negated.

- **Redefine Function** defines a wrapper for a random non-attribute function, and replaces a call to the original function with a call to the wrapper.

A subsequent validation pass permits only code which terminates and returns a value within 5 seconds without er-

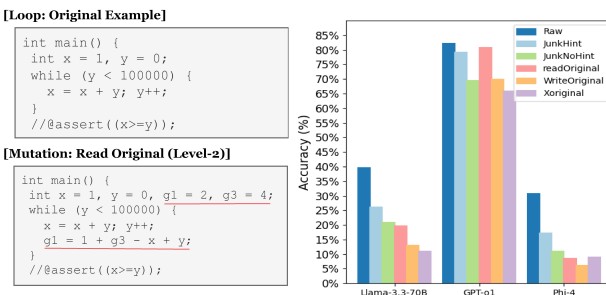

Figure 6. **Left**: An example from Loop dataset and its *Read Original* mutation. **Right**: Loop concise results – accuracy of the best-performing model in each family. Full results can be found in Figure 25 of Appendix H.

rors. Code transformations for CRUXEval are deterministic, since for any validated program, the output of the code on the provided inputs is taken to be ground truth. See Appendix G for additional implementation detail.

### 5.2. Symbolic Mutations to Loop ②

Unlike earlier benchmarks, applying substantial mutations to Loop tasks is more challenging. After changing the values of the program variables, loop invariants can cease to exist. Hence, we limit ourselves to **Level-2** mutations that add irrelevant information in the form of additional variables and operations, leaving the values of variables from the original program unaffected. Table 7 of Appendix H describes these mutations, summarized below.

- **Junk Hint** adds five new variables junk_0, junk_1, etc., to the program (Si et al., 2018). Before the loop, they are initialized with randomly generated constants. Within the loop body, each new variable is updated with arithmetic expressions over randomly generated constants and the new variables.

- **Junk No-Hint** assigns names to the new variables that resemble those in the original program. This aims to prevent LLMs from identifying variables with junk in their name as unnecessary.

- **Read Original** reads the original variables of the code into the newly introduced ones. Original variables and operators are randomly sampled and added to new variables.

- **Write Original** introduces superficial additional writes to the original variables, i.e. they leave the values taken by these variables at runtime unaffected. For a new variable $y$, existing variables are incremented by an identically zero polynomial $f(y)$. To further obscure the underlying identity, polynomial coefficients are decomposed and rearranged.

- **X-Original** applies both **Read** and **Write Original**.

Although the mutated programs are syntactically larger, all mutations described above have the property that the loop

invariant of the original program is also a valid loop invariant for the mutated program.

### 5.3. Model Performance

Zero-shot accuracy on CRUXEval tasks are summarized in Figure 5. [5] Overall, accuracy scores decrease for both **Level-2** and **Level-3** mutations compared with factual problems, with **Level-3** posing the greatest challenge. The drop in performance on **Level-2** mutations is a particularly strong indication of memorization effects from benchmark leakage, as these mutations intentionally introduce minimal change to the execution complexity and corresponding code understanding skills. Notably, the original CRUXEval benchmark is publicly available and LLM-generated, both of which provide potential leakage pathways.

Evaluations on Loop tasks are summarized in Figure 6 (for evaluation with all models, see Figure 25 in Appendix H). Smaller models show performance degradation on introducing junk named variables. The performance decline from "*Junk Hint*" to "*Junk No-Hint*" shows that LLMs use variable names as semantic cues. The "*X-Original*" mutations confuse all models and significantly degrades their success rates compared to the original tasks.

## 6. Ablation Study and Discussion

We further evaluate the models' reasoning abilities through two ablation experiments: (1) to determine whether the performance drop on mutated questions stems from the mutations themselves or from the additional reasoning complexity introduced by some mutations; and (2) to assess the impact of in-context learning examples on the models' reasoning performance.

### 6.1. The Influence of Reasoning Complexity

We use GSM8K as a testbed for our analysis. Following Ye et al. (2025), we quantify the complexity of a numerical reasoning question by the number of reasoning steps, defined as the number of operations in its code solution. By this measure, all mutated questions introduce one additional reasoning step compared to their original counterparts, except for those in the *SampleValues* category. Notably, while *UselessInfo* questions also include an extra reasoning step, this step does not affect the final answer, so we still categorize *UselessInfo* as a *Level-2* mutation. Using this definition, we compute the average accuracy of each model across examples with different numbers of reasoning steps. Given the large number of models tested, we aggregate their results

---

[5]Each mutation type applies only to a subset of the questions in CRUXEval, so we also evaluate performance on each mutation with respect to the corresponding (factual) subset of problems (see Figure 24 in Appendix G).

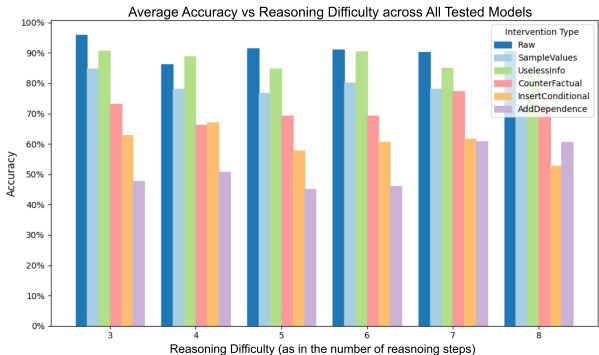

*Figure 7.* We analyze model performance across varying numbers of reasoning steps in the GSM8K benchmark. The x-axis represents the number of reasoning steps, defined as the number of operations in the code solution. The plot aggregates the performance of all tested models. Detailed results for each model are provided in Figure 15 in Appendix E.5.

and report the overall average accuracy in Figure 7. Detailed per-model results are in Figure 15 in Appendix E.5.

From Figure 7, we draw two key observations: (1) even when controlling for the number of reasoning steps, models consistently perform worse on mutated questions than on the original test set; and (2) *Level-3* mutations pose a substantially greater challenge than *Level-2*. Notably, the accuracy on *Level-3* mutations with just three reasoning steps is significantly lower than on original test examples requiring seven reasoning steps. These findings suggest that the primary cause of performance degradation on mutated questions is the mutations themselves, rather than the added reasoning complexity.

### 6.2. The impact of in-context examples

Following previous studies, the 8-shot in-context examples used in the GSM8K experiments in Section 4 are randomly sampled from the original training set. Here, we further investigate whether including mutated examples in the in-context examples can improve model performance on the generated test set variations. As shown in Figure 8, we find that most models achieve significantly higher accuracy on the test variations when given both original and mutated examples, compared to using only original GSM8K examples or only mutated examples as in-context examples. A detailed explanation of the experiment, including additional results and full prompt examples, is in Appendix E.2.

## 7. Conclusion

This work presents a novel framework, RE-IMAGINE, designed to assess the reasoning capabilities of LLMs through the systematic generation of challenging problem variations. Our findings reveal a consistent decline in model performance as reasoning complexity increases across all evaluated benchmarks. On our mutated GSM8K benchmark, we

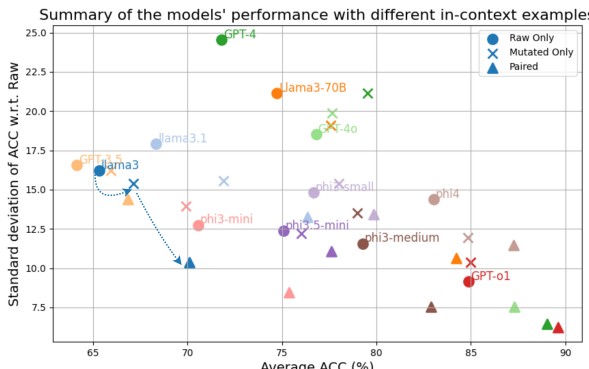

*Figure 8.* Models' performance when prompted with different types of in-context examples. We summarize model performance across two dimensions: the average accuracy across all mutated test sets and the original test set (x-axis), and the standard deviation in accuracy on the five mutated test sets w.r.t. the raw test set (y-axis). The bottom-right corner reflects the ideal balance of high performance on both raw and mutated sets. In this plot, dots indicate models prompted only with original examples, crosses represent models prompted only with mutated examples, and triangles denote models prompted with both original and corresponding mutated examples. Model structures are differentiated by color, with the same model represented by the same color.

find that the overall performance of all LLMs degrades with increasing ladder levels. The o1 model shows greater robustness on this benchmark and achieves the best performance among all tested models on bi-counterfactuals. However, even the most powerful models struggle with structured problem variations, particularly when these variations are combined. Similar results were observed on CLadder.

Additionally, we successfully applied RE-IMAGINE to mutate code benchmarks Loop and CruxEval. Despite the mutations being designed to have minimal effect on code length, execution time, and overall question difficulty, a substantial decline in performance is again observed across models and problem variations. This lends additional evidence that these models struggle with higher levels of the reasoning hierarchy, suggesting that gaps in their capabilities may be hidden by effects such as memorization and benchmark leakage.

In addition to the models discussed in previous sections, we report results for other popular models in Appendix D. The conclusions are found to generalize well to these models.

RE-IMAGINE provides a systematic framework to assess and expose these weaknesses, highlighting the need for more rigorous evaluation strategies. Expanding RE-IMAGINE to broader domains could further enhance our understanding of reasoning capabilities and limitations of LLMs. Our hope is for RE-IMAGINE to provide a foundational framework for a more nuanced evaluation of LLMs and encourage the development of more robust reasoning models.

## Impact Statement

The RE-IMAGINE framework represents a significant advancement in evaluating the reasoning capabilities of Large Language Models. By systematically generating problem variations that cannot be solved through memorization alone, RE-IMAGINE provides a robust method to disentangle genuine reasoning from statistical recall. The findings reveal notable drops in LLM performance when faced with problem variations. By setting a new standard for evaluating hierarchical reasoning, this research paves the way for more transparent and interpretable AI systems, ultimately contributing to the broader goal of building models with true generalizable reasoning capabilities. This will impact the development of future AI systems. It will also influence benchmark design, ensuring that evaluations better reflect genuine cognitive capabilities rather than memorization. Additionally, this research can guide policymakers, researchers, and practitioners in designing more reliable and trustworthy AI applications across domains such as education, healthcare, and scientific discovery, where robust reasoning is critical.

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

## A. Appendix: Language Model Details

| Models | Details | #Parameters | Pre-train Size (tokens) |
|---|---|---|---|
| Phi-3 mini | *microsoft/Phi-3-mini-128k-instruct* | 3.8B | 4.9T |
| Phi-3.5 mini | *microsoft/Phi-3.5-mini-instruct* | 3.8B | 3.4T |
| Phi-3 small | *microsoft/Phi-3-small-128k-instruct* | 7B | 4.8T |
| Phi-3 medium | *microsoft/Phi-3-medium-128k-instruct* | 14B | 4.8T |
| Phi-4 | *microsoft/phi-4* | 14B | 9.8T |
| Llama 2 | *meta-llama/Llama-2-7b-chat-hf* | 7B | 2T |
| Llama 3 | *meta-llama/Meta-Llama-3-8B-Instruct* | 8B | 15T |
| Llama 3.1 | *meta-llama/Meta-Llama-3.1-8B-Instruct* | 8B | 15T |
| Llama 3.3 (70B) | *meta-llama/Meta-Llama-3.3-70B-Instruct* | 8B | 15T |
| Llama 3 (70B) | *meta-llama/Meta-Llama-3-70B-Instruct* | 70B | 15T |
| GPT-3.5 | *gpt-35-turbo_1106* | 175B | – |
| GPT-4 | *gpt-4-32k_0613* | – | – |
| GPT-4o | *gpt-4o_2024-08-06* | – | – |
| GPT-o1 | *o1-preview_2024-09-12* | – | – |

*Table 3.* Details of LLMs used in this work.

## B. Appendix: Comparison to Related Work

| Previous Studies | Mutations | Manual Effort | Scalability |
|---|---|---|---|
| GSM-IC (Shi et al., 2023) | UselessInfo (L2) | Hand-written patterns/rules for each question | Scaling within questions |
| iGSM (Ye et al., 2025) | Out of scope constructed from scratch | Human-defined concepts, dependency graphs, NL expression patterns for QA pairs. | Scaling within questions |
| GSM-Symbolic (Mirzadeh et al., 2024) | UselessInfo (L2) SampleValues (L2) ChangingVariableNames (L2) | Hand-written patterns/rules for each question | Scaling within questions |
| GSM-Hard (Gao et al., 2023) | SampleValues (L2) | Writing prompts to get symbolic representation. Writing program to sample values for parameters. | Scaling within benchmarks, since the code generation replies on the ground truth CoT. |
| CounterFactual (Hüyük et al., 2024) | Bi-Counterfactual (L3) | Hand-written patterns/rules for each question | Scaling within questions |
| Re-Imagine (Ours) | Covering L2 and L3 | Customizing reusable adapters | Scaling across benchmarks |

*Table 4.* Compared to existing work. Scaling within questions means that for each original question with human-written patterns, an arbitrary number of mutated questions can be generated. Scaling within benchmarks means that the mutation generation tool could help to generate an arbitrary number of mutated questions for all questions in the entire benchmark.

## C. Appendix: Fundamental Link to Causality

The three levels of Pearl's ladder of causation – intervention, correlation and counterfactuals – provide a foundational hierarchical characterization for the reasoning levels of an AI system. Our work is inspired by this ladder, and taking the three levels as a reference, we describe a hierarchical set of experiments that also capture different levels of reasoning in LLMs. The key connection between both frameworks is the problem's computation graph, which can be understood as a causal model in Pearl's framework. Each problem in a benchmark can be interpreted as a single realization of the graph with specific node values. Experiments associated with different perturbations in such a graph can often be related to operations in Pearl's ladder of causation. For instance, computing the effect of a change to one leaf node maps to the standard definition of a counterfactual from causal inference. Note, however, that not all mutations in the three levels have a causal counterpart (for example, adding an irrelevant piece of information or changing an operation). In this sense, our framework covers a broader scope of reasoning tasks in each level.

# D. Appendix: Extra Models Performance on GSM8K, CruxEval and Loop

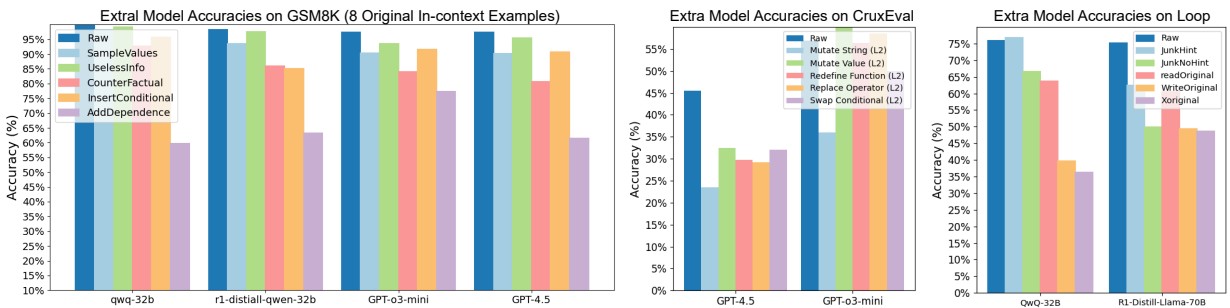

*Figure 9.* In addition to the models discussed in the main body of the paper, we also report results for several popular models on GSM8K, CruxEval, and Loop. The conclusions drawn in the main paper are found to generalize well to these additional models.

# E. Appendix: GSM8K Details

## E.1. Statistical Accuracy on GSM8K and its variations

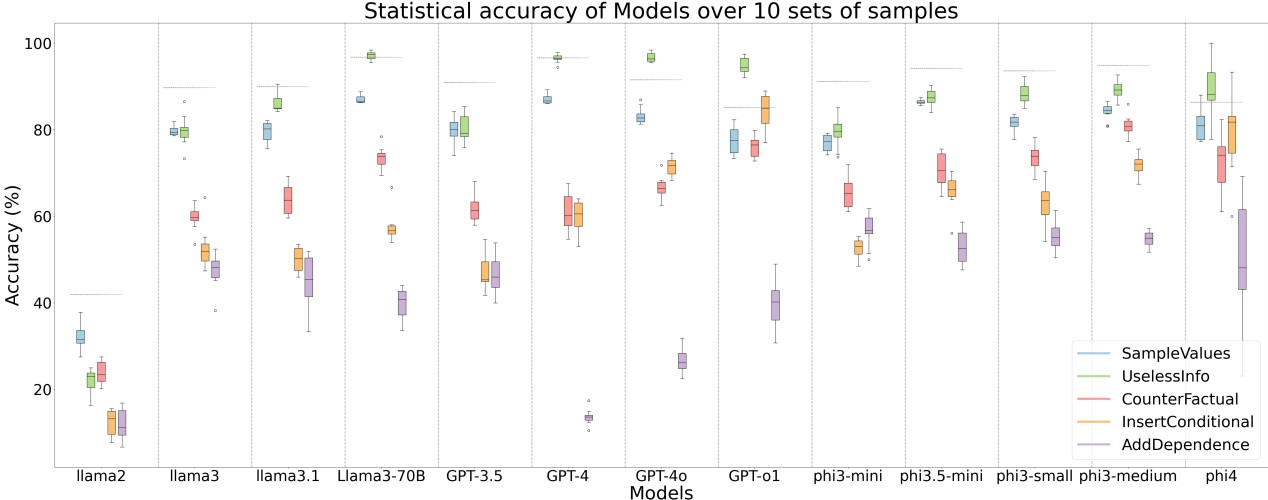

*Figure 10.* The **statistical accuracy** of models on GSM8K numerical answer predictions is evaluated across different mutated test sets at varying reasoning levels (see Section 3.2 and Figure 1). To generate this plot, we randomly select 150 examples from the pool of examples eligible for all mutations, excluding *Bi-CounterFactual*. We then create 10 test variations for each mutation by sampling with 10 different seeds. All models are independently tested on these 10 variations, and the resulting accuracies are used to generate the bar plot. During testing, each model is prompted with 8 raw examples along with their Chain-of-Thought (CoT) process. Our observations are as follows: **(1)** All models show significant accuracy variability when tested on mutated test sets, particularly on variations with *Level-3* mutations; **(2)** For all models, the average performance on the mutated sets is notably lower than on the original GSM8K test set (as indicated by the dashed line). Moreover, for most test set variations with *Level-3* mutations, even the accuracy on the most beneficial sample set falls below the accuracy of the original GSM8K test. This suggests that a single sentence mutation altering the computational logic of the original math question may severely impact the models' performance.

## E.2. The Influence of In-context Examples

In this section, we examine the impact of in-context examples on the model's reasoning ability.

First, we aim to investigate whether providing the model with only mutated examples can improve its reasoning performance on the generated test set variations. During testing, all models are provided with seven mutated examples with amended CoT process, each from a different type of mutation (including two additional mutations beyond those discussed in Section 3.2 and Figure 1). Figure 19 shows one detailed prompt. Figure 11 presents the results, indicating that the models exhibit

behavior highly similar to when original GSM8K examples are used as in-context examples (see Figure 2).

Next, we investigate whether providing the model with both the original example and its mutated variations can enhance its reasoning performance on the generated test set variations. During testing, all models receive seven in-context examples. Each example consists of an original GSM8K question and its answer, followed by a mutated version of the question along with the corresponding CoT solution and answer. These examples cover different types of mutations, including two additional mutations beyond those discussed in Section 3.2 and Figure 1. To maintain consistency with the in-context example format, we also include the original question and its answer for the final mutated question that the model is expected to answer. Figure 20 illustrates a detailed example of the prompt, while Figure 12 presents the results. The findings indicate that ***most models perform significantly better on the generated test set variations when provided with both original and mutated examples***, compared to using only original GSM8K examples (Figure 2) or only mutated examples (Figure 19) as in-context examples.

To visualize the comparison, we summarize model performance across two dimensions: the average accuracy across all six test sets, as in the left bar plot (x-axis), and the standard deviation in accuracy on the five mutated test sets w.r.t. the raw test set (y-axis). The bottom-right corner reflects the ideal balance of high performance on both raw and mutated sets. In this way, we consolidate model performance across these three different in-context example scenarios in Figure 8 and illustrate how to interpret the plot using the Llama3 model as an example.

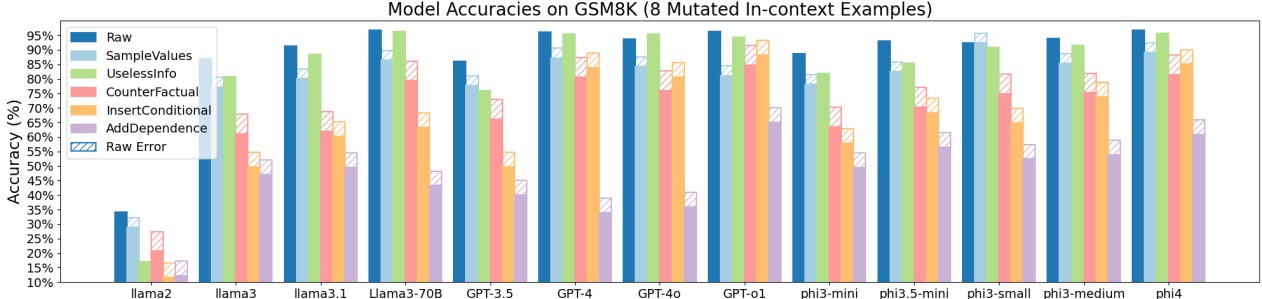

*Figure 11.* GSM8K: Detailed accuracy for all models in the *Phi*, *Llama*, and *GPT* families. All models are prompted with **7 mutated in-context examples**. Check Figure 19 for the detailed prompt. Due to potential noise from the automatic mutation process, we performed a human evaluation on the mutated test set. We added the percentage of invalid examples in each mutation category to the top of each accuracy bar, assuming that if these QA pairs were correct, the models would answer them correctly. This provides an upper bound on the model's performance.

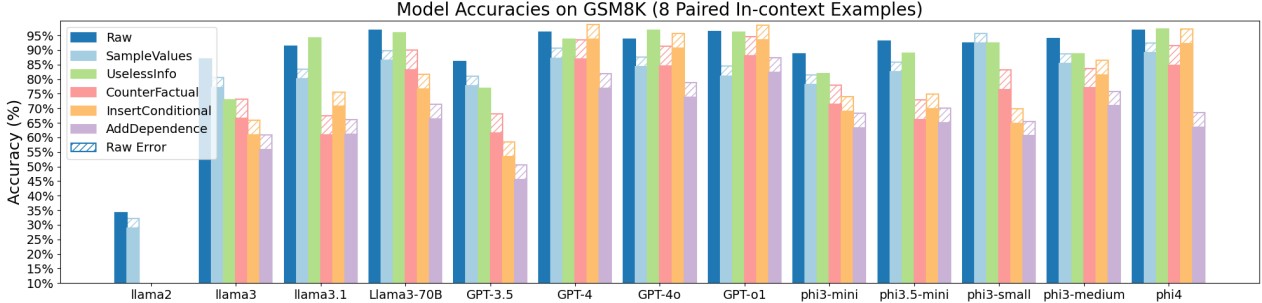

*Figure 12.* GSM8K: Detailed accuracy for all models in the *Phi*, *Llama*, and *GPT* families. All models are prompted with **7 paired raw and the corresponding mutated in-context examples**. Check Figure 20 for the detailed prompt. The missing Llama-2 experiments are due to its failure to follow instructions and produce correctly formatted answers.

### E.3. Composition of Mutations

The defined *Ladder* of reasoning hierarchy (Section 2) and the automatic variation generation pipeline (Section 3), open the possibility of combining multiple mutations to create challenging test set variations. In this section, we combine the *SampleValues* mutation from *Level-2* with the remaining four mutations (excluding *Bi-CounterFactual*) to create four test set variations, each containing two types of mutations. The models' accuracy can be find in Figure 13.

To visualize the comparison, similar to Figure 8 in Section 6.2, we summarize model performance along two dimensions: the average accuracy across all test sets and the standard deviation in accuracy on the mutated test sets relative to the raw test set. We combine model performance on test set variations with a single mutation and those with two mutations in Figure 14. As observed, all models exhibit lower average accuracy on test variations containing two mutations compared to those with a single mutation. The increased standard deviation of accuracy w.r.t. the original test set suggests that *composing mutations expands the performance gap between the mutated sets and the original set*.

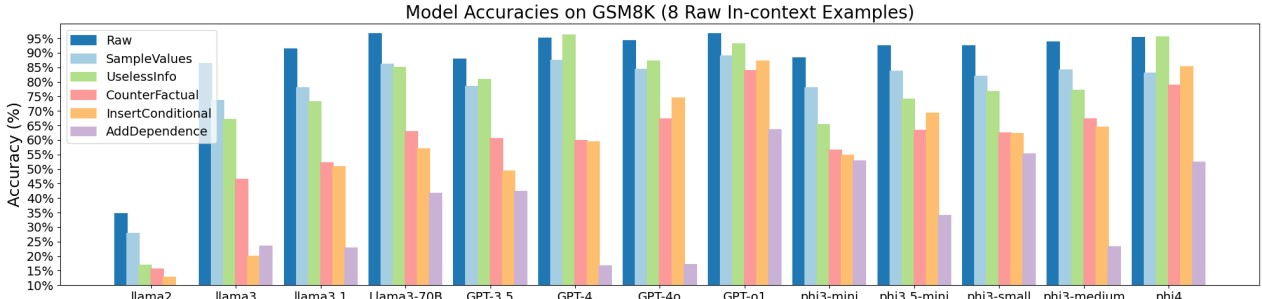

*Figure 13.* GSM8K: Detailed accuracy for all models in the *Phi*, *Llama*, and *GPT* families. All models are prompted with **8 original in-context examples**. However, with the exception of *SampleValues*, all other mutations are combined with *SampleValues* to create even more challenging test set variations.

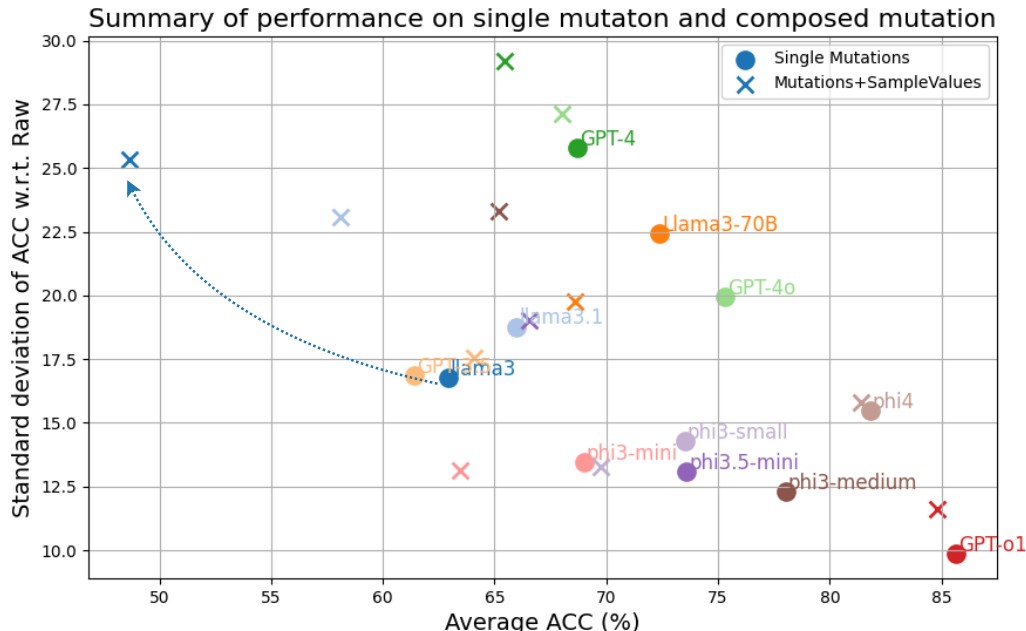

*Figure 14.* GSM8K: Visualization comparing model performance on test set variations with one or two mutations. We summarize model performance across two dimensions: the average accuracy across all six test sets, as in the left bar plot (x-axis), and the standard deviation in accuracy on the five mutated test sets w.r.t. the raw test set (y-axis). The bottom-right corner reflects the ideal balance of high performance on both raw and mutated sets. In this plot, dots represent models' accuracy on test sets with a single mutation, while crosses indicate their accuracy on sets containing two mutations. Model structures are differentiated by color, with the same structure represented by the same color.

## E.4. Decomposing Model Performance by Number of Reasoning Steps

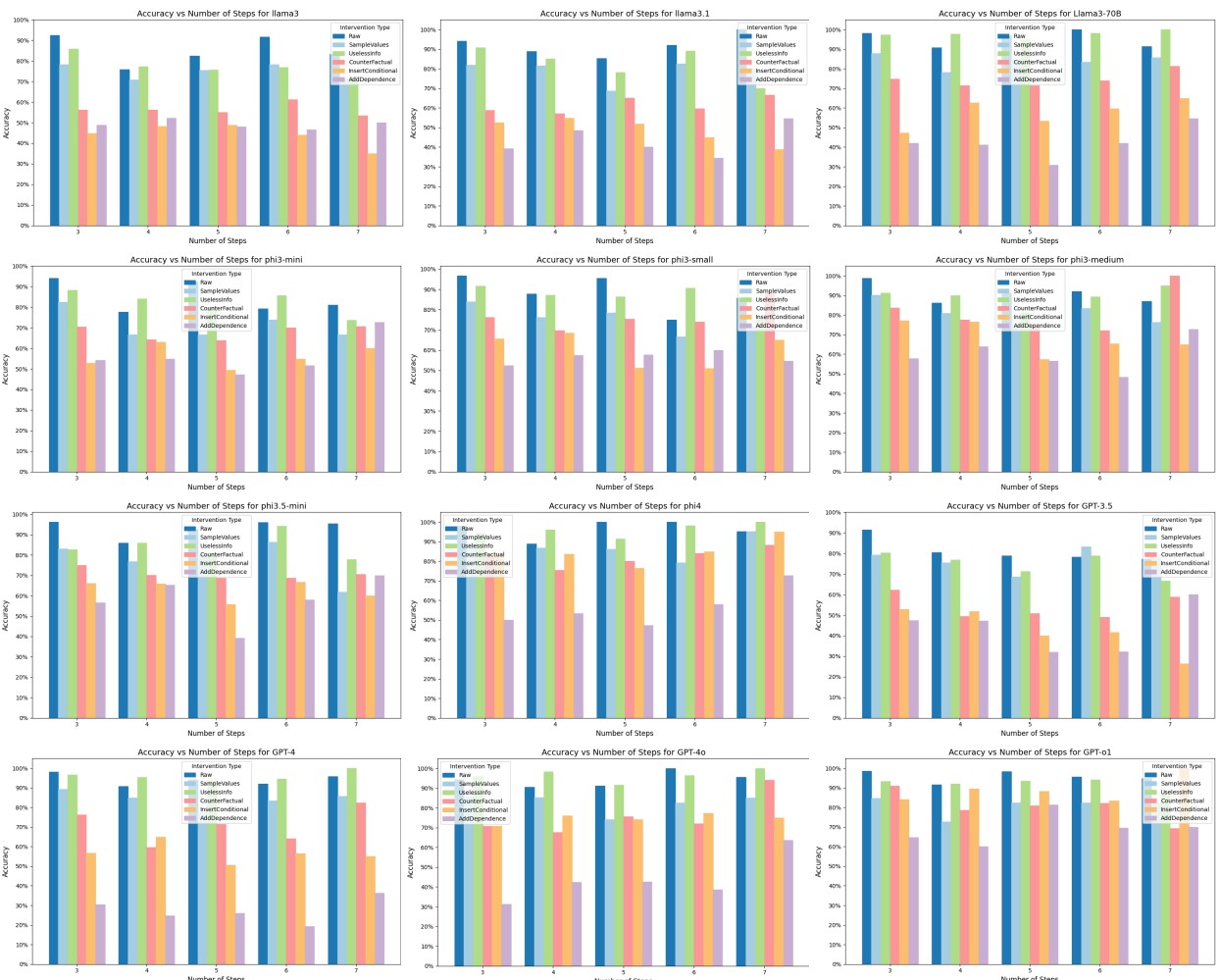

*Figure 15.* GSM8K: We analyze model performance across varying numbers of reasoning steps. We define the difficulty of a numerical reasoning question by the number of calculation steps (operations) in its code solution. For each model, we compute the average accuracy across examples grouped by their number of calculation steps. In the plots, the x-axis indicates the number of reasoning steps, and the y-axis shows the accuracy. The results reveal that, across all models, even when controlling for the number of reasoning steps, performance is consistently lower on mutated examples than on the original test set. Furthermore, Level-3 mutations pose a significantly greater challenge than Level-2. Notably, models perform worse on Level-3 mutations with just three calculation steps than they do on original test examples requiring seven steps, indicating a substantial gap.

## E.5. Detailed Prompts

```
[{"role": "system", "content": "I will provide a math question along with a sample of Python code which
                                 solves the question. The names of variables in this code may be intentionally
                                 misleading. Then I will provide an updated version of the math question which
                                 differs from the original question. It may include irrelevant information or
                                 references to variables which are not used in the code. I would like you to
                                 update the code to represent the inclusion of this additional sentence, so
                                 that it answers the updated version of the math question accurately. You may
                                 need to add or remove variables or lines of code. It is important that you
                                 disregard the names of the variables in the code and focus on the values they
                                 represent. Do not modify which variable is returned at the end of the code. If
                                 you are unable to complete the task, please return the phrase
                                 'Unable to complete'."},
 {"role": "user", "content": "Math question: {original_text}\n\n"
                              "Python code: {original_code}\n\n"
                              "Updated math question: {mutated_text}\n\n"
                              "Requested output format: Runnable python code with no additional text."
                              "Do not include `python at the beginning.")}]
```

*Figure 16.* Prompts used in GSM8K quality control. We prompt the LLM to back-translate the mutated math problem into Python by modifying the original question's Python solution. The generated code must produce an execution result that matches the ground truth answer of the mutated question.

```
[{"role": "system", "content": "You are a helpful assistant."},
 {"role": "user", "content": "I will provide a math question along with the Python code that solves it.
                              I will set a particular variable in the code to a different value.
                              I would like you to describe this change in natural language
                              and offer a brief explanation for it in a single sentence.
                              Math question: {original_question}\n\n
                              Python code: {original_code}\n\n
                              I set {variable_name} to {new_value}.
                              Describe this change in natural language.
                              The request format is: The change is that ... instead of... ."}}]
```
**OverWriteValue**

```
[{"role": "system", "content": "I will provide a math question along with the Python code that solves it.
                              I will add one new line of code which describes the definition of a new,
                              irrelevant variable. I would like you to create a meaning for this
                              new variable which is related to the setting of the word problem,
                              but ultimately irrelevant to the question. The change to the word problem
                              should be a single sentence in natural language, describing the new
                              variable and its definition, which makes sense in context.
                              The change must not affect the rest of the word problem. Do not make
                              any reference to the fact that the new information is irrelevant,
                              but ensure that it is. Do not use the name of the variable from the
                              python code; only describe the change in natural language."},
 {"role": "user", "content": "Math question: {original_question}\n\n
                              Original Python code: {original_code}\n\n
                              I add the line {new_line} which defines the variable {new_variable}.\n\n
                              Describe this change in natural language.
                              The request format is: \"Suppose that ... .\""}}]
```
**UselessInfo**

```
[{"role": "system", "content": "I will provide a math question along with the Python code that solves it.
                              I will define a new variable, and then add the value of that new variable
                              to the definition of an existing variable. I would like you to describe
                              an alternate version of the word problem which matches the updated code.
                              The change to the word problem should be a single sentence in natural
                              language, which makes sense in context. Make sure to explicitly state the
                              value of the new variable and how it is used in the problem."},
 {"role": "user", "content": "Math question: {original_question}\n\n
                              Original Python code: {original_code}\n\n
                              I define the new variable {v2} to be: {v2_new_value}
                              I update the definition of variable {v1} to: {v1_value}.\n\n
                              Describe this change in natural language.
                              The request format is: \"Suppose that ... in addition to ... .\""}}]
```
**AddDependence**

```
[{"role": "system", "content": "I will provide a math question along with the Python code that solves it.
                              I will add an if-else conditional to the code. I would like you to
                              describe an alternate version of the word problem whose solution matches
                              the updated code. The change to the word problem should be a single
                              sentence in natural language, without ambiguity, and which makes sense
                              in context."},
 {"role": "user", "content": "Math question: {original_question}\n\n
                              Original Python code: {original_code}\n\n
                              I replace the line {original_line} with the conditional statement
                              {conditional_replacement}.\n\n
                              Describe this change in natural language.
                              The request format is: Suppose that ... if ... otherwise ... ."}}]
```
**InsertConditional**

*Figure 17.* Prompts used in GSM8K mutated symbolic representation to natural language.

```
[
    {
        "role": "user",
        "content": "Question: Mimi picked up 2 dozen seashells on the beach.  Kyle found twice as many
                    shells as Mimi and put them in his pocket. Leigh grabbed one-third of the shells
                    that Kyle found.  How many seashells did Leigh have?"
    },
    {
        "role": "assistant",
        "content": "Answer: Mimi has 2 x 12 = <<2*12=24>>24 sea shells.\n
                    Kyle has 24 x 2 = <<24*2=48>>48 sea shells.\n
                    Leigh has 48 / 3 = <<48/3=16>>16 sea shells.\n#### 16"
    },
    {
        "role": "user",
        "content": "Question: Frankie's parents let him have many pets. He has six more snakes than he
                    has cats. He has one less parrot than cats. Six of his pets have four legs. He has
                    2 dogs. How many pets does he have in total?"
    },
    {
        "role": "assistant",
        "content": "Answer: He has 6 - 2 = <<6-2=4>>4 cats.\n
                    He has 4 - 1 = <<4-1=3>>3 parrots.\n
                    He has 4 + 6 = <<4+6=10>>10 snakes.\n
                    He has a total of 2 + 4 + 3 + 10 = <<2+4+3+10=19>>19 pets.\n#### 19"
    },
    ... 6 more examples
    {
        "role": "user",
        "content": "Question: In mid-May , the depth of a river in Moreland is measured . By mid-June ,
                    the river is 17 feet deeper than mid-May . By mid-July , the river is 3 times deeper
                    than mid-June . If the river is 54 feet deep in mid-July , how many feet deep was
                    the river in mid-May ? Let's think step by step. At the end, you MUST write the answer
                    after '####'. Note that, this might be a trick question designed to confuse LLMs with
                    additional information. Look for irrelevant information, new assumptions or distractors
                    in the question. "
    }
]
```

*Figure 18.* Prompts used in GSM8K testing. The prompt contains 8 raw in-context examples.

```
[
    {
        "role": "user",
        "content": "Question: If Janet reads 80 pages a day and Belinda reads 30 pages a day, assume Belinda actually reads 50
pages a day, how many more pages does Janet read in 6 days?"
    },
    {
        "role": "assistant",
        "content": "Answer: First find the number of pages Belinda actually reads per day: 50 pages/day. Then find the total
difference in the number of pages Janet and Belinda read each day: 80 pages/day - 50 pages/day = <<80-50=50>>30 pages/day Then
multiply the daily difference by the number of days to find the total difference in the number of pages they read: 30 pages/day *
6 days = <<30*6=180>>180 pages #### 180"
    },
    {
        "role": "user",
        "content": "Question: If Janet reads 80 pages a day and Belinda reads 30 pages a day. Suppose that Belinda only reads if
Janet reads more than 100 pages a day; if she reads 100 pages a day, Belinda does not read at all. How many more pages does Janet
read in 6 days?"
    },
    {
        "role": "assistant",
        "content": "Answer: First find the number of pages Belinda actually reads per day: since Janet reads 80 pages per day
which is less than 100 pages, Belinda reads 0 pages/day. Then find the total difference in the number of pages Janet and Belinda
read each day: 80 pages/day - 0 pages/day = <<80-0=80>>80 pages/day Then multiply the daily difference by the number of days to
find the total difference in the number of pages they read: 80 pages/day * 6 days = <<80*6=480>>480 pages #### 480"
    },
    {
        "role": "user",
        "content": "Question: Lee mows one lawn and charges $33. Last week he mowed 16 lawns and three customers each gave him a
$10 tip. Assume Lee needs to pay the three customers $10 each since he damaged their lawns instead of getting $10 tips. How many
dollars did Lee earn mowing lawns last week?"
    },
    {
        "role": "assistant",
        "content": "Answer: 33 * 16 = $<<33*16=528>>528 3 * 10 = $<<3*10=30>>30 528 - 30 = $<<528-30=498>>498 Lee earned $498
mowing lawns last week. #### 498"
    },
    {
        "role": "user",
        "content": "Question: Lee mows one lawn and charges $33. Last week he mowed 16 lawns and three customers each gave him a
$10 tip. Assume Lee's neighbor also mowed 16 lawns last week. How many dollars did Lee earn mowing lawns last week?"
    },
    {
        "role": "assistant",
        "content": "Answer: 33 * 16 = $<<33*16=528>>528 3 * 10 = $<<3*10=30>>30 528 + 30 = $<<528+30=558>>558 Lee earned $558
mowing lawns last week. #### 558"
    },
    {
        "role": "user",
        "content": "Question: Lee mows one lawn and charges $33. Last week he mowed 16 lawns and three customers each gave him a
$10 tip. Assume Lee got payed in pounds not dollars. How many pounds did Lee earn mowing lawns last week?"
    },
    {
        "role": "assistant",
        "content": "Answer: 33 * 16 = $<<33*16=528>>528 3 * 10 = $<<3*10=30>>30 528 + 30 = $<<528+30=558>>558 Lee earned $558
mowing lawns last week. #### 558"
    },
    {
        "role": "user",
        "content": "Question: Lee mows one lawn and charges $33. Last week he mowed 16 lawns and three customers each gave him a
$10 tip. Assume Lee got extra bonus from his company $5 for each customer he served. How many dollars did Lee earn mowing lawns
last week?"
    },
    {
        "role": "assistant",
        "content": "Answer: 33 * 16 = $<<33*16=528>>528 3 * 10 = $<<3*10=30>>30 3 * 5 = $<<3*5=15>>15 528 + 30 + 15 =
$<<528+30+15=573>>573 Lee earned $573 mowing lawns last week. #### 573"
    },
    {
        "role": "user",
        "content": "Question: Lee mows one lawn and charges $33. Last week he mowed 16 lawns and three customers each gave him a
$10 tip. Assume Lee did not get tips from his customers. How many dollars did Lee earn mowing lawns last week?"
    },
    {
        "role": "assistant",
        "content": "Answer: 33 * 16 = $<<33*16=528>>528 Lee earned $528 mowing lawns last week. #### 528"
    },
    {
        "role": "user",
        "content": "Question: Bill had to finish a project from work that was to take him 4 days. He took 6 seven-hour naps in
the four days. Assume the total time spent on naps is set to 38 hours instead of being calculated as the product of the number of
naps and the duration of each nap. How long did he spend working on the project? Let's think step by step. At the end, you MUST
write the answer after '####'. Note that, this might be a trick question designed to confuse LLMs with additional information.
Look for irrelevant information, new assumptions or distractors in the question. "
    }
]
```

*Figure 19.* Prompts used in GSM8K testing. The prompt contains 7 in-context examples demonstrating different types of mutations.

```
// preamble or system instruction
As an expert problem solver, solve step by step a mutated math question based on the original question and its answer.
Note that, this might be a trick question designed to confuse LLMs with additional information.
Look for irrelevant information, new assumptions or distractors in the question.
At the end, you MUST write the answer after '####'.

// shot-1
The original question: If Janet reads 80 pages a day and Belinda reads 30 pages a day, how many more pages does Janet read in 6 days?
The answer of the question above is: 300
The mutated question: If Janet reads 80 pages a day and Belinda reads 30 pages a day, assume Belinda actually reads 50 pages a day, how many more pages does
Janet read in 6 days?
The answer of the question above is: First find the number of pages Belinda actually reads per day: 50 pages/day. Then find the total difference in the number of
pages Janet and Belinda read each day: 80 pages/day - 50 pages/day = <<80-50=50>>30 pages/day Then multiply the daily difference by the number of days to find
the total difference in the number of pages they read: 30 pages/day * 6 days = <<30*6=180>>180 pages #### 180

// shot-2
The original question: If Janet reads 80 pages a day and Belinda reads 30 pages a day, how many more pages does Janet read in 6 days?
The answer of the question above is: 300
The mutated question: If Janet reads 80 pages a day and Belinda reads 30 pages a day. Suppose that Belinda only reads if Janet reads more than 100 pages a day;
if she reads 100 pages a day, Belinda does not read at all. How many more pages does Janet read in 6 days?
The answer of the question above is: First find the number of pages Belinda actually reads per day: since Janet reads 80 pages per day which is less than 100
pages, Belinda reads 0 pages/day. Then find the total difference in the number of pages Janet and Belinda read each day: 80 pages/day - 0 pages/day = <<80-
0=80>>80 pages/day Then multiply the daily difference by the number of days to find the total difference in the number of pages they read: 80 pages/day * 6 days
= <<80*6=480>>480 pages #### 480

// shot-3
The original question: Lee mows one lawn and charges $33. Last week he mowed 16 lawns and three customers each gave him a $10 tip. How many dollars did Lee earn
mowing lawns last week?
The answer of the question above is: 558
The mutated question: Lee mows one lawn and charges $33. Last week he mowed 16 lawns and three customers each gave him a $10 tip. Assume Lee needs to pay the
three customers $10 each since he damaged their lawns instead of getting $10 tips. How many dollars did Lee earn mowing lawns last week?
The answer of the question above is: 33 * 16 = $<<33*16=528>>528 3 * 10 = $<<3*10=30>>30 528 - 30 = $<<528-30=498>>498 Lee earned $498 mowing lawns last week.
#### 498

// shot-4
The original question: Lee mows one lawn and charges $33. Last week he mowed 16 lawns and three customers each gave him a $10 tip. How many dollars did Lee earn
mowing lawns last week?
The answer of the question above is: 558
The mutated question: Lee mows one lawn and charges $33. Last week he mowed 16 lawns and three customers each gave him a $10 tip. Assume Lee's neighbor also
mowed 16 lawns last week. How many dollars did Lee earn mowing lawns last week?
The answer of the question above is: 33 * 16 = $<<33*16=528>>528 3 * 10 = $<<3*10=30>>30 528 + 30 = $<<528+30=558>>558 Lee earned $558 mowing lawns last week.
#### 558

// shot-5
The original question: Lee mows one lawn and charges $33. Last week he mowed 16 lawns and three customers each gave him a $10 tip. How many dollars did Lee earn
mowing lawns last week?
The answer of the question above is: 558
The mutated question: Lee mows one lawn and charges $33. Last week he mowed 16 lawns and three customers each gave him a $10 tip. Assume Lee got payed in pounds
not dollars. How many pounds did Lee earn mowing lawns last week?
The answer of the question above is: 33 * 16 = $<<33*16=528>>528 3 * 10 = $<<3*10=30>>30 528 + 30 = $<<528+30=558>>558 Lee earned $558 mowing lawns last week.
#### 558

// shot-6
The original question: Lee mows one lawn and charges $33. Last week he mowed 16 lawns and three customers each gave him a $10 tip. How many dollars did Lee earn
mowing lawns last week?
The answer of the question above is: 558
The mutated question: Lee mows one lawn and charges $33. Last week he mowed 16 lawns and three customers each gave him a $10 tip. Assume Lee got extra bonus from
his company $5 for each customer he served. How many dollars did Lee earn mowing lawns last week?
The answer of the question above is: 33 * 16 = $<<33*16=528>>528 3 * 10 = $<<3*10=30>>30 3 * 5 = $<<3*5=15>>15 528 + 30 + 15 = $<<528+30+15=573>>573 Lee earned
$573 mowing lawns last week. #### 573

// shot-7
The original question: Lee mows one lawn and charges $33. Last week he mowed 16 lawns and three customers each gave him a $10 tip. How many dollars did Lee earn
mowing lawns last week?
The answer of the question above is: 558
The mutated question: Lee mows one lawn and charges $33. Last week he mowed 16 lawns and three customers each gave him a $10 tip. Assume Lee did not get tips
from his customers. How many dollars did Lee earn mowing lawns last week?
The answer of the question above is: 33 * 16 = $<<33*16=528>>528 Lee earned $528 mowing lawns last week. #### 528

// target question
The original question: On a quiz, Martin answered three fewer questions correctly than Kelsey, and Kelsey answered eight more questions correctly than Campbell.
Campbell answered 35 questions correctly. How many did martin answer correctly?
The answer of the question above is: 40
The mutated question: On a quiz, Martin answered three fewer questions correctly than Kelsey, and Kelsey answered eight more questions correctly than Campbell.
Campbell answered 35 questions correctly. Assume Kelsey answered seven more questions correctly than Campbell instead of eight more questions. How many did
martin answer correctly?
The answer of the question above is:
```

*Figure 20.* Prompts used in GSM8K testing. The prompt contains 7 in-context examples. Each example contains a raw question and answer, with the corresponding mutated question and chain-of-thought fo the mutated question, with the final answer. The 7 examples are demonstrating different types of mutations.

# F. Appendix: CLadder Details

## F.1. Benchmark Details

The CLadder dataset (Jin et al., 2023b) provides a systematic evaluation of causal reasoning abilities in LLMs. It consists of 10,000 *causal graphs* of binary variables (i.e., Bernoulli conditional distributions) that encompass common treatment-effect estimation scenarios, such as confounding, mediation, and collisions. Each causal graph is associated with multiple *queries*, spanning the three levels of Pearl's ladder of causation: associational, interventional, and counterfactual. Each dataset example includes (1) a *causal graph*, (2) a *query*, (3) a *causal engine* that computes over the casual graph and the query to get a binary answer, and (4) a template-based formulation that translates the causal graph and the query into a natural language *question* for the LLM to interface with. The benchmark evaluates models' causal reasoning abilities by requiring them to (1) extract causal concepts from natural language and (2) apply causal inference (either implicitly or explicitly) over the causal concepts, such as do-calculus, to calculate the final answer.

## F.2. Pipeline Details

### F.2.1. QUESTION TO SYMBOLIC REPRESENTATION ①

**Filtering:** We begin with a filtering pass of the 10,000 examples in CLadder, removing all examples corresponding to query types 'backdoor adjustment' and 'collider bias' (in total 1,747 examples). The former was filtered due to the inherent complexity of the python it would require, and the latter because it is only present for one type of causal graph structure. This leaves 8,365 examples from CLadder after filtering.

**Parser:** Next, we develop a parser to automatically translate these examples into *Python code* with the help of the *causal engine* provided by CLadder. The parser determines the query type from the meta data in the CLadder examples, and extracts the relevant variables from the natural language questions. For every query type, causal graph, and estimand, the parser generates a snippet of executable Python code which computes the estimand.

**Validation:** To check that the NL to Python code parser is successful, after conversion of the examples, we execute the Python code and compare the computed estimand value with its ground-truth in CLadder. 60 examples showed a discrepancy, leaving us with 8,305 examples (99.34% coverage of parsed examples).

### F.2.2. SYMBOLIC MUTATIONS ② AND MUTATED SYMBOLIC REP TO NL ③

To examine the robustness of LLMs in handling variations in question phrasing and irrelevant information, we include two key mutations:

- *UselessInfo*: We add extraneous information at two levels. (i) *RelatedIrrelevantInfo*: With Meta-Llama 70B Instruct, we generate two sentences for each *question* that are semantically related but causally irrelevant. (ii) *UnrelatedIrrelevantInfo*: We prepend the natural language description from a randomly selected CLadder example to the beginning of each *question*, ensuring that the added sentences remain semantically irrelevant to prevent ambiguity. Both *RelatedIrrelevantInfo* and *UnrelatedIrrelevantInfo* describe **Level-2** mutations.

- *CounterFactual*: We modify the probability values in the original *question* by introducing a counterfactual assumption, such as: "Suppose the probability of `<Event>` is `<new-value>` instead.". To generate the ground-truth answer for the counterfactual question, we update the corresponding probability of the conditional distribution in *Python code* and execute it. *CounterFactual* is a **Level-3** mutation.

In Table 5, we provide an example for each type of mutation. Additionally, we generate two more test set variations by combining the *CounterFactual* mutation with each *UselessInfo* mutation. The full results are shown in Figure 21.

| Mutation type | Problem and Bayesian Network | Probability distributions | Code |
|---|---|---|---|
| Original | For husbands that don't set the alarm and wives that don't set the alarm, the probability of ringing alarm is 8%. For husbands that don't set the alarm and wives that set the alarm, the probability of ringing alarm is 54%. For husbands that set the alarm and wives that don't set the alarm, the probability of ringing alarm is 41%. For husbands that set the alarm and wives that set the alarm, the probability of ringing alarm is 86%. For husbands that don't set the alarm, the probability of alarm set by wife is 74%. For husbands that set the alarm, the probability of alarm set by wife is 24%. *If we disregard the mediation effect through wife, would husband positively affect alarm clock?* 

 $H$ (husband sets alarm) 
 $W$ (wife sets alarm) 
 $A$ (alarm rings) 

 $P(A=1\|H=0,W=0)=.08$ 
 $P(A=1\|H=0,W=1)=.54$ 
 $P(A=1\|H=1,W=0)=.41$ 
 $P(A=1\|H=0,W=0)=86$ 
 $P(W=1\|H=0)=.74$ 
 $P(W=1\|H=1)=.24$ 

 Estimand: 

 $\mathbb{E}[Y_{X=1,V2=0} - Y_{X=0,V2=0}]$ 
 $= \sum_{V2=v} P(V2=v\|X=0) \cdot$ 
 $[P(Y=1\|X=1,V2=v)$ 
 $- P(Y=1\|X=0,V2=v)]$ | ```
# Probabilities
p_a_given_not_h_not_w = 0.08
p_not_a_given_not_h_not_w = 1 - p_a_given_not_h_not_w
p_a_given_not_h_w = 0.54
p_not_a_given_not_h_w = 1 - p_a_given_not_h_w
p_a_given_h_not_w = 0.41
p_not_a_given_h_not_w = 1 - p_a_given_h_not_w
p_a_given_h_w = 0.86
p_not_a_given_h_w = 1 - p_a_given_h_w
p_w_given_not_h = 0.74
p_not_w_given_not_h = 1 - p_w_given_not_h
p_w_given_h = 0.24
p_not_w_given_h = 1 - p_w_given_h

# For W = 0
term_0 = p_not_w_given_not_h * (p_a_given_h_not_w -
    p_a_given_not_h_not_w)

# For W = 1
term_1 = p_w_given_not_h * (p_a_given_h_w -
    p_a_given_not_h_w)

# Final sum
result = term_0 + term_1
if result > 0:
    print("yes")
else:
    print("no")
``` |
| CounterFactual | For husbands that don't set the alarm and wives that don't set the alarm [...] the probability of alarm set by wife is 24%. *Assume that the probability of the alarm ringing, if both the husband and the wife don't set the alarm, changes to 25%. If we disregard the mediation effect through wife, would husband positively affect alarm clock?* | $\mathbf{P(A=1\|H=0,W=0)=.25}$ 
 $P(A=1\|H=0,W=1)=.54$ 
 $P(A=1\|H=1,W=0)=.41$ 
 $P(A=1\|H=0,W=0)=86$ 
 $P(W=1\|H=0)=.74$ 
 $P(W=1\|H=1)=.24$ | ```
# Probabilities
p_a_given_not_h_not_w = 0.25
p_not_a_given_not_h_not_w = 1 - p_a_given_not_h_not_w
p_a_given_not_h_w = 0.54
p_not_a_given_not_h_w = 1 - p_a_given_not_h_w
p_a_given_h_not_w = 0.41
p_not_a_given_h_not_w = 1 - p_a_given_h_not_w
p_a_given_h_w = 0.86
p_not_a_given_h_w = 1 - p_a_given_h_w
p_w_given_not_h = 0.74
p_not_w_given_not_h = 1 - p_w_given_not_h
p_w_given_h = 0.24
p_not_w_given_h = 1 - p_w_given_h

# For W = 0
term_0 = p_not_w_given_not_h * (p_a_given_h_not_w -
    p_a_given_not_h_not_w)

# For W = 1
term_1 = p_w_given_not_h * (p_a_given_h_w -
    p_a_given_not_h_w)

# Final sum
result = term_0 + term_1
if result > 0:
    print("yes")
else:
    print("no")
``` |
| IrrelevantInfo | For husbands that don't set the alarm and wives that don't set the alarm [...] the probability of alarm set by wife is 24%. *Assume that the probability of the kids going to school each day is 80%.* If we disregard the mediation effect through wife, would husband positively affect alarm clock? | $K$ (kids to school) 

 $P(A=1\|H=0,W=0)=.25$ 
 $P(A=1\|H=0,W=1)=.54$ 
 $P(A=1\|H=1,W=0)=.41$ 
 $P(A=1\|H=0,W=0)=86$ 
 $P(W=1\|H=0)=.74$ 
 $P(W=1\|H=1)=.24$ | (identical to original) |

*Table 5.* Mutations in RE-IMAGINE for the CLadder dataset. In the Bayesian network, gray nodes indicate a variable whose probability mass function (conditional on variables from incoming edges) has been intervened on or added as part of the mutation. The probability distributions for the irrelevant graph in IrrelevantInfo are not stated. Note that the probability of the positive event specifies the full conditional distribution since all distributions are Bernoulli (e.g. $P(A=1\|H=0,W=0)=0.08 \implies P(A=0\|H=0,W=0)=0.92$).

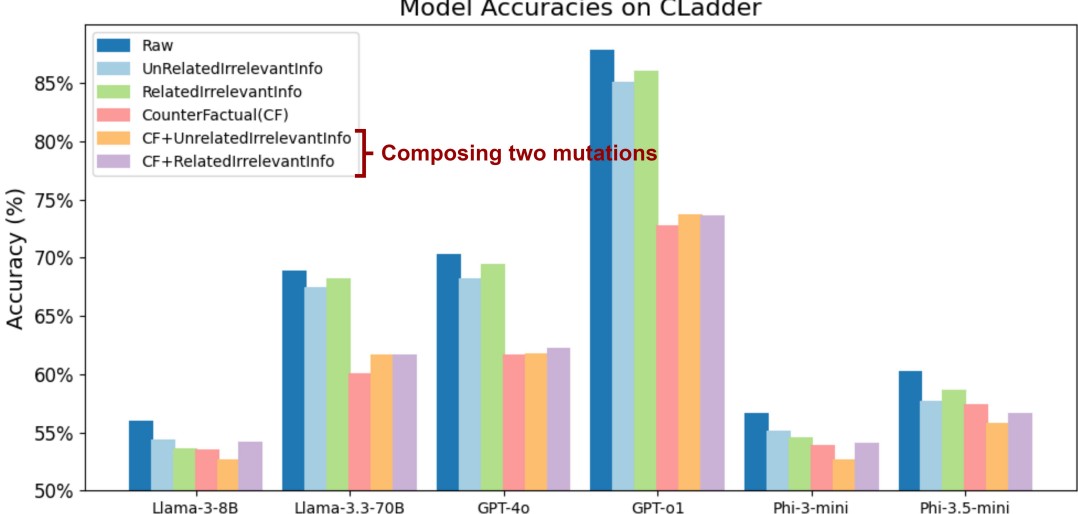

*Figure 21.* CLadder: Detailed accuracy for models in the *Phi*, *Llama*, and *GPT* families.

### F.3. Prompts for mutated questions

Below are illustration of NL prompts for our mutations on CLadder. We highlight parts of the original question in red, and the mutated natural language in blue. The prompt consists of the System prompt (stated first below), followed by a mutation-specific prompt (stated thereafter).

**System Prompt** We use the following system prompt throughout the evaluation.

```
You are an expert at causal inference and reasoning.  You will be given a question and you must answer with
"yes" or "no" only.
```

**Related UselessInfo**

```
Imagine a self-contained, hypothetical world with only the following conditions, and without any unmentioned
factors or causal relationships:  The probability of the husband being awake when the wife sets the alarm is
85%.  If the husband sets the alarm, the probability of the wife being in a good mood is 73%.  Husband has a
direct effect on wife and alarm clock.  Wife has a direct effect on alarm clock.
For husbands that don't set the alarm and wives that don't set the alarm, the probability of ringing alarm is
11%.  For husbands that don't set the alarm and wives that set the alarm, the probability of ringing alarm is
60%.  For husbands that set the alarm and wives that don't set the alarm, the probability of ringing alarm is
46%.  For husbands that set the alarm and wives that set the alarm, the probability of ringing alarm is 92%.
For husbands that don't set the alarm, the probability of alarm set by wife is 61%.  For husbands that set
the alarm, the probability of alarm set by wife is 1%.
Does husband positively affect alarm clock through wife?
```

**Unrelated UselessInfo**

```
Imagine a self-contained, hypothetical world with only the following conditions, and without any unmentioned
factors or causal relationships:  The man in the room has a direct effect on room.  The candle has a direct
effect on room.The overall probability of blowing out the candle is 68%.  The probability of not blowing out
the candle and dark room is 12%.  The probability of blowing out the candle and dark room is 51%.  Imagine a
self-contained, hypothetical world with only the following conditions, and without any unmentioned factors
or causal relationships:  Husband has a direct effect on wife and alarm clock.  Wife has a direct effect on
alarm clock.
For husbands that don't set the alarm and wives that don't set the alarm, the probability of ringing alarm is
8%.  For husbands that don't set the alarm and wives that set the alarm, the probability of ringing alarm is
54%.  For husbands that set the alarm and wives that don't set the alarm, the probability of ringing alarm is
41%.  For husbands that set the alarm and wives that set the alarm, the probability of ringing alarm is 86%.
For husbands that don't set the alarm, the probability of alarm set by wife is 74%.  For husbands that set
the alarm, the probability of alarm set by wife is 24%.
If we disregard the mediation effect through wife, would husband positively affect alarm clock?"
```

**CounterFactual**

```
Imagine a self-contained, hypothetical world with only the following conditions, and without any unmentioned
factors or causal relationships:  Husband has a direct effect on wife and alarm clock.  Wife has a direct
effect on alarm clock.
For husbands that don't set the alarm and wives that don't set the alarm, the probability of ringing alarm is
11%.  For husbands that don't set the alarm and wives that set the alarm, the probability of ringing alarm is
60%.  For husbands that set the alarm and wives that don't set the alarm, the probability of ringing alarm is
46%.  For husbands that set the alarm and wives that set the alarm, the probability of ringing alarm is 92%.
For husbands that don't set the alarm, the probability of alarm set by wife is 61%.  For husbands that set
the alarm, the probability of alarm set by wife is 1%.
Suppose Probability of ringing alarm, given that alarm did not set by husband alarm set by wife is 0.46.
Does husband positively affect alarm clock through wife?
```

## Unrelated UselessInfo x CounterFactual

```
Imagine a self-contained, hypothetical world with only the following conditions, and without any unmentioned
factors or causal relationships:  The man in the room has a direct effect on room.  The candle has a direct
effect on room.The overall probability of blowing out the candle is 68%.  The probability of not blowing out
the candle and dark room is 12%.  The probability of blowing out the candle and dark room is 51%.  Imagine a
self-contained, hypothetical world with only the following conditions, and without any unmentioned factors
or causal relationships:  Husband has a direct effect on wife and alarm clock.  Wife has a direct effect on
alarm clock.
For husbands that don't set the alarm and wives that don't set the alarm, the probability of ringing alarm is
11%.  For husbands that don't set the alarm and wives that set the alarm, the probability of ringing alarm is
60%.  For husbands that set the alarm and wives that don't set the alarm, the probability of ringing alarm is
46%.  For husbands that set the alarm and wives that set the alarm, the probability of ringing alarm is 92%.
For husbands that don't set the alarm, the probability of alarm set by wife is 61%.  For husbands that set
the alarm, the probability of alarm set by wife is 1%.
Suppose Probability of ringing alarm, given that alarm did not set by husband alarm set by wife is 0.46.
Does husband positively affect alarm clock through wife?
```

## Related UselessInfo x CounterFactual

```
Imagine a self-contained, hypothetical world with only the following conditions, and without any unmentioned
factors or causal relationships:  The probability of the husband being awake when the wife sets the alarm is
85%.  If the husband sets the alarm, the probability of the wife being in a good mood is 73%.   Husband has
a direct effect on wife and alarm clock.  Wife has a direct effect on alarm clock.
For husbands that don't set the alarm and wives that don't set the alarm, the probability of ringing alarm is
11%.  For husbands that don't set the alarm and wives that set the alarm, the probability of ringing alarm is
60%.  For husbands that set the alarm and wives that don't set the alarm, the probability of ringing alarm is
46%.  For husbands that set the alarm and wives that set the alarm, the probability of ringing alarm is 92%.
For husbands that don't set the alarm, the probability of alarm set by wife is 61%.  For husbands that set
the alarm, the probability of alarm set by wife is 1%.
Suppose Probability of ringing alarm, given that alarm did not set by husband alarm set by wife is 0.46.
Does husband positively affect alarm clock through wife?
```

## F.4. Prompts for generating IrrelevantInfo mutations

To generate the NL question for the IrrelevantInfo mutations, we use the following auxiliary prompt. The template
<question> refers to the part highlighted in red in the prompts stated in Appendix F.3.

### System Prompt to Generate Irrelevant Information

```
You are tasked with generating irrelevant probability statements to enhance causal reasoning questions.
These statements will be added to the beginning of the question, right after the phrase:  "Imagine a
self-contained, hypothetical world with only the following conditions, and without any unmentioned factors
or causal relationships:" The irrelevant probability statements must adhere to the following guidelines:
They should describe probabilities or conditional probabilities related to entities, actions, or settings
described in the question.  The probabilities should be realistic and plausible but should have no impact on
the causal reasoning task or relationships in the question.  They must blend seamlessly into the hypothetical
context without introducing new causal relationships.  They should add complexity to the question but not
distract from solving the core problem.  Structure of Irrelevant Probability Statements:  Use probabilities
or percentages (e.g., "The probability of X is Y%").  Include conditional probabilities where appropriate
(e.g., "If A occurs, the probability of B is C%").  Ensure the statements align with the general tone
of hypothetical worlds while remaining inconsequential to the reasoning process.  Your Task:  Generate
two irrelevant probability statement for a given question.  Ensure they are consistent with the context,
plausible, and add complexity without affecting the causal reasoning.  Focus on adhering to the structure and
requirements outlined above.  You must ONLY give the two statements as output and nothing else.  DO NOT start
with phrases like "Here are two irrelevant probability statements for the given question:" <question>
```

# G. Appendix: CRUXEval Details

## G.1. Mutations

We summarize the mutation types implemented for CRUXEval in Table 6. Note that each of the five code mutations can be implemented as both a *Type-2* and a *Type-3* mutation, depending on how the resulting question is posed; the corresponding two prompt templates are presented in Appendix G.2.

| Mutation type | Description | Original Code | Mutated Code |
|---|---|---|---|
| Replace Operator | Selects a random operator of type `ast.BinOp`, `ast.UnaryOp`, `ast.BoolOp`, or `ast.AugAssign` for replacement. Replaces basic arithmetic operators with their inverses (e.g. addition with subtraction). Flips boolean operators and unary operators (e.g. "and" with "or"). Replaces comparison operators with their negations (eg. "in" with "not in" and "≤" with ">"). | ```python
def f(text, lower, upper):
  count = 0
  new_text = list()
  for char in text:
    char = lower if char.isdecimal()
    ↪  else upper
    if char in ['p', 'C']:
      count += 1
    new_text.append(char)
  return count, ''.join(new_text)
``` | ```python
def f(text, lower, upper):
  count = 0
  new_text = list()
  for char in text:
    char = lower if char.isdecimal()
    ↪  else upper
    if char not in ['p', 'C']:
      count += 1
    new_text.append(char)
  return count, ''.join(new_text)
``` |
| Mutate String | Selects a random string instance for replacement. Replaces the string with a random sequence of the same length. | ```python
def f(text, lower, upper):
  count = 0
  new_text = list()
  for char in text:
    char = lower if char.isdecimal()
    ↪  else upper
    if char in ['p', 'C']:
      count += 1
    new_text.append(char)
  return count, ''.join(new_text)
``` | ```python
def f(text, lower, upper):
  count = 0
  new_text = list()
  for char in text:
    char = lower if char.isdecimal()
    ↪  else upper
    if char in ['E', 'C']:
      count += 1
    new_text.append(char)
  return count, ''.join(new_text)
``` |
| Mutate Value | Selects a random instance of type `bool`, `int`, or `float` for replacement. Boolean values are replaced with their negations; integers are perturbed by a uniformly random nonzero integer between -10 and 10; floats are perturbed by a uniformly random nonzero float between -10 and 10. | ```python
def f(nums):
  output = []
  for n in nums:
    output.append((nums.count(n), n))
  output.sort(reverse=True)
  return output
``` | ```python
def f(nums):
  output = []
  for n in nums:
    output.append((nums.count(n), n))
  output.sort(reverse=False)
  return output
``` |
| Swap Conditional | Selects a random conditional node for replacement. If both an `if` and `else` branch are present, the code body for each branch is swapped. If only an `if` branch is present, the condition of the branch is negated (if X becomes `if not X`). | ```python
def f(t):
  for c in t:
    if not c.isnumeric():
      return False
    else:
      return True
``` | ```python
def f(t):
  for c in t:
    if not c.isnumeric():
      return True
    else:
      return False
``` |
| Redefine Function | Selects a random function call for replacement (attribute function calls are not included). Defines a new wrapper function which calls the original function, and replaces the original function call with a call to the new function. | ```python
def f(dic):
  for k,v in sorted(dic.items(),
  ↪  key=lambda x:
  ↪  len(str(x))[:-1]:
    dic.pop(k)
  return list(dic.items())
``` | ```python
def xxxz(arg0):
  return list(arg0)

def f(dic):
  for k, v in sorted(dic.items(),
  ↪  key=lambda x:
  ↪  len(str(x))[:-1]:
    dic.pop(k)
  return xxxz(dic.items())
``` |

*Table 6.* Mutations in RE-IMAGINE for the CruxEval dataset.

## G.2. Prompts

Below we list the prompts used in testing CRUXEval. All models are tested using zero-shot prompting, only.

Prompt for *Level-2* mutations:

```
Consider the following code with a missing value represented by '??':    {f_question}

Based on the given Python code, which may contain errors, complete the assert statement with the output
when executing the code on the given test case.  Print only the exact text to replace "??" in the assert
statement, to make the assert statement true.  Do NOT output any extra information, even if the function is
incorrect or incomplete.
```

Prompt for *Level-3* mutations:

```
Consider the following code with a missing value represented by '??':    {f_question}

Suppose a change is now made to the code, as described by the following diff: {diff}

Based on the given Python code, which may contain errors, complete the assert statement with the output

when executing the code on the given test case.  Print only the exact text to replace "??" in the assert

statement, to make the assert statement true.  Do NOT output any extra information, even if the function is

incorrect or incomplete.
```

## G.3. Coverage

Not all transformations are applicable to all problems; we report coverage statistics below. Mutations are performed on 800 total factual examples, with 88.9% of factual examples covered by at least one mutation.

| Mutate String | Mutate Value | Redefine Function | Replace Operator | Swap Conditional |
|---|---|---|---|---|
| 30.3% | 56.6% | 54.1% | 50.6% | 42.5% |

*Figure 22.* Mutation Coverage Statistics (as a percentage of total CRUXEval data)

## G.4. Evaluation and Matched Factual Accuracy

We plot the accuracy on the factual and mutated CRUXEval benchmark for ten language models in Figure 21.

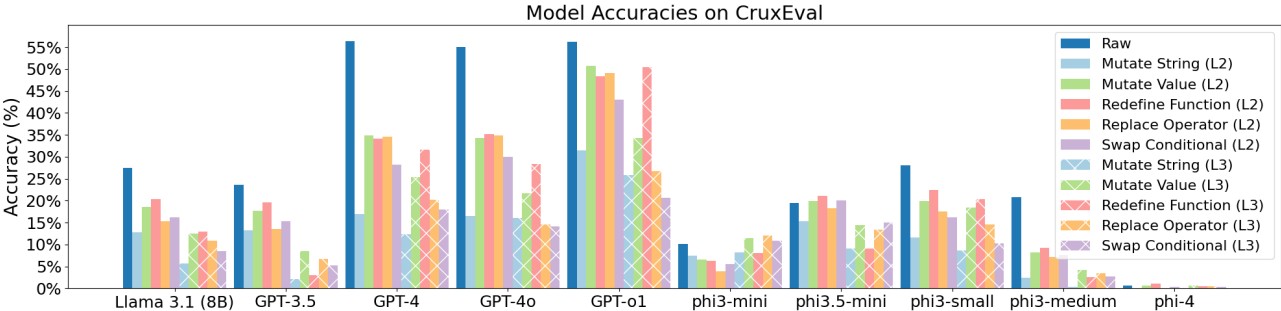

*Figure 23.* CRUXEval: Detailed accuracy for models in the *Phi*, *Llama*, and *GPT* families.

Because the coverage statistics for each mutation fall well below 100%, each mutation is tested on only a subset of the total CRUXEval benchmark problems. We therefore consider the possibility that there may be a correlation between *which mutations apply* to a given problem and the *difficulty of that problem* for the models we evaluate. In order to account for this fact, we also compute *matched factual accuracy scores* for each mutation. These scores report the factual accuracy rate of the model when tested on only the same subset of problems to which the mutation applies. For example, in Figure 24, the green bars represent accuracy scores on the factual, *Level-2*, and *Level-3* variants of the 56.6% of CRUXEval problems which admit a Mutate Value mutation. Note for ease of reading that the palest bar of each color in Figure 24 represents the matched factual score for the corresponding mutation.

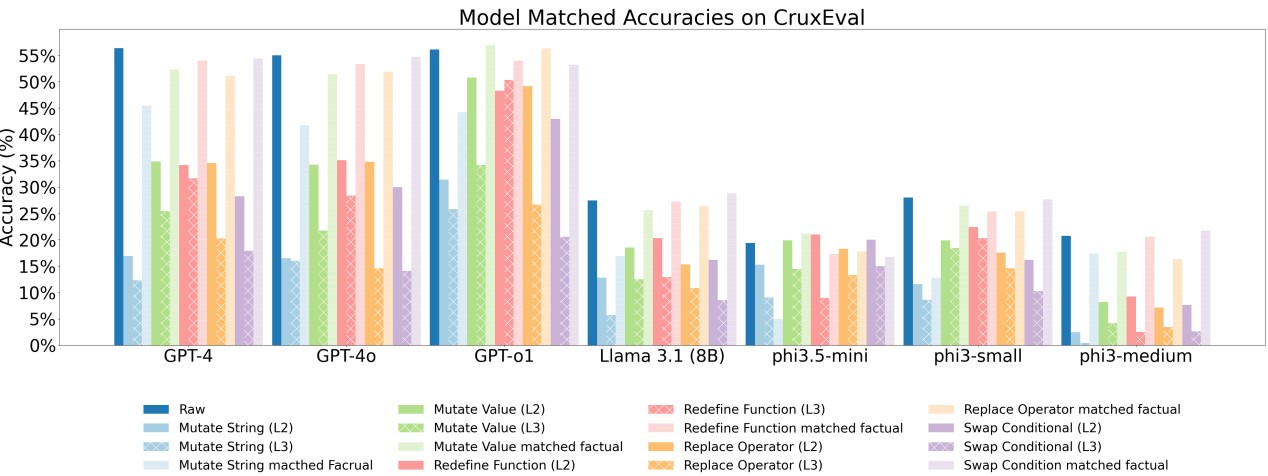

*Figure 24.* CRUXEval: Detailed accuracy and *matched factual accuracy* for models in the *Phi*, *Llama*, and *GPT* families.

We observe that the overall trend of decreasing accuracy with **Level-2** and **Level-3** mutations is still evident with respect to matched accuracy scores. Nonetheless, we note that matched factual accuracy is consistently lower than raw accuracy on certain mutations; most drastically for Mutate String. This indicates that problems which contain mutable strings tend to be more difficult for LLMs than other CRUXEval problems, underscoring the importance of viewing matched scores for the fullest picture of model performance. The further decrease between matched factual scores and the corresponding **Level-2** and **Level-3** mutation scores indicates that the difficulty-level correlation cannot account for the remaining loss in accuracy post-mutation.

## H. Appendix: Loop Details

In software verification, automatic inference of loop invariants is a classic problem (Si et al., 2018). Since this problem is undecidable in general, many heuristics have been proposed, including those based on machine learning. Recently, LLMs have been demonstrated to perform well on loop invariant inference of integer programs (Wu et al., 2024; Kamath et al., 2024). In this paper, we mutate such tasks and evaluate the efficacy of LLMs on the mutated tasks.

Each task has a program with a loop and an assertion. The goal is to infer a predicate that satisfies the following three conditions: it holds before the loop starts executing, holds for each iteration of the loop, and implies the assertion when the loop exits. Finding such predicates can be tricky even for small programs. Consider the task in Figure 6; the predicate $x \geq y$ satisfies the first and the third condition but fails to satisfy the second. As another example, consider the simpler program

```
x=0; while (x< 100) x++; assert x==100
```

Given this program, the goal is for an LLM to produce the loop invariant $x \leq 100$. It is easy to see that $x \leq 100$ satisfies all the three conditions specified above. Note that there are infinitely many predicates that are variations of $x \leq 100$, e.g., $x \leq 99 \lor x = 100$, that are valid loop invariants and the model succeeds if it infers any of them. Although inferring such loop invariants is undecidable, in practice, checking whether a given candidate invariant satisfies the three conditions can be done well by automated software verification tools like Frama-C (Correnson et al.).

To evaluate a model on an original or a mutated task, the LLM output is checked by Frama-C that internally uses SMT solvers such as Z3(de Moura & Bjørner, 2008), alt-ergo(Ait El Hara et al., 2022) and CVC4(Barrett et al., 2011). The model succeeds on the task if Frama-C succeeds to verify the LLM output as a loop invariant, and fails otherwise. Hence, a model can fail on a task for two reasons: either Frama-C declares the candidate as invalid, or the candidate invariant is valid but Frama-C could not prove its validity within the provided time bound. We evaluate on 250 tasks from Kamath et al. (2024) which have the property that Frama-C succeeds in verifying that the invariant for the original program is also the invariant for all the mutated programs.

Unlike GSM8K, applying *Level-3* mutations to these tasks is difficult. Once we change the values of the program variables, loop invariants can cease to exist. For example, if we mutate the example above then we can get the mutated program

```
x=101; while (x< 100) x++; assert x==100
```

with a new initialization. There is no loop invariant which discharges the assertion as the assertion will get violated when the program is run. Hence, we limit ourselves to a category of *Level-2* mutations that add useless information to the tasks in the form of additional variables and operations that leave the values of the variables in the original program unaffected.

Table 7 shows how various mutations operate on the program in Figure 6. Figure 25 shows the results of all the models we consider on 250 loop invariant inference tasks from Kamath et al. (2024). We use the following prompt from (Kamath et al., 2024), reproduced here for completeness.

```
You are a helpful AI software assistant that reasons about how code behaves.  Given a program, you can find
loop invariants, which can then be used to verify some property in the program.
Frama-C is a software verification tool for C programs.  The input to Frama-C is a C program file with ACSL
(ANSI/ISO C Specification Language) annotations.
For the given program, find the necessary loop invariants of the while loop to help Frama-C verify the
post-condition.

Instructions:
- Make a note of the pre-conditions or variable assignments in the program.
- Analyze the loop body and make a note of the loop condition.
- Output loop invariants that are true
(i) before the loop execution,
(ii) in every iteration of the loop and
(iii) after the loop termination,
such that the loop invariants imply the post condition.
- If a loop invariant is a conjunction, split it into its parts.
- Output all the loop invariants in one code block.  For example:
```
/*@
loop invariant i1;
loop invariant i2;
*/
```
Rules:
- **Do not use variables or functions that are not declared in the program.**
- **Do not make any assumptions about functions whose definitions are not given.**
- **All undefined variables contain garbage values.  Do not use variables that have garbage values.**
- **Do not use keywords that are not supported in ACSL annotations for loops.**
- **Variables that are not explicitly initialized, could have garbage values.  Do not make any assumptions
about such values.**
- **Do not use the
at(x, Pre) notation for any variable x.**
- **Do not use non-deterministic function calls.**
Consider the following C program:
```
code
```
You are allowed to use implication to take care of the conditional nature of the code.  Use implication (⇒)
instead of using if-then.

For all variables, add conjunctions that bound the maximum and minimum values that they can take, if such
bounds exist.

If a variable is always equal to or smaller or larger than another variable, add a conjunction for their
relation.

If the assertion is guarded by a condition, use the guard condition in an implication.

If certain variables are non-deterministic at the beginning or end of the loop, use an implication to make
the invariant trivially true at that location.

Output the loop invariants for the loop in the program above.  Let's think step by step.
```

| Mutation | Description | Code | Intervention Categories |
|---|---|---|---|
| Vanilla | Base Snippet with a loop and assertion. | ```int x = 1, y = 0;
while (y < 100000) {x+=y; y++;}
//@ assert(x >= y);``` | None |
| Junk Hint | Adding 2 junk variables that are disjoint from original variables and are named as junk_0 and junk_1. | ```int x = 1, y = 0;
int junk_0 = 1, junk_1 = 3;
while (y < 100000) {x+=y; y++;
  junk_0 = 178; junk_1 = junk_0 - 687;
}//@ assert(x >= y);``` | Add irrelevant Info |
| Junk No-Hint | Adding 2 new variables that are disjoint from original variables and are named unremarkably (g0 and g1). Create new statements using randomly sampling of new variables and constants. | ```int x = 1, y = 0;
int g0 = 1, g1 = 3;
while (y < 100000) { x+=y; y++;
  g0 = 178; g1 = g0 - 687;
}//@ assert(x >= y);``` | Add irrelevant Info and rename nodes |
| Read Original | Read the original variables of the code into the newly introduced ones. Randomly sample original variables and operators and add them to new variables. The arithmetic expressions in updates to new variables can use the original program variables along with the new variables and random constants. This mutation introduces more reads of the original program variables but no writes to them. | ```int x = 1, y = 0;
int g0 = 1, g1 = 3;
while (y < 100000) { x+=y; y++;
  g0 = 178  + x - y; g1 = g0 - 687 + y - x;
}//@ assert(x >= y);``` | Add irrelevant Info and rename nodes, dummy relationship between nodes |
| Write Original | Increment the original variables by algebraic identities of new variables that equate to 0. Randomly sample new variables for the identity. Split constants to increase complexity. | ```int x = 1, y = 0;
int g0 = 1, g1 = 3;
while (y < 100000) {
x = x + y + (((g0 + g1)*(g0 + g1)) -1*g0*g1 -2*g0*g1)
    - ((g0*g0 + g1*g1) - 1*g0*g1);
y = y + 1 + ((g1*g1 + g0*g0) -1*g1*g0) - (((g1 + g0)*(
    g1 + g0)) -1*g1*g0 -2*g1*g0);
g0 = 178; g1 = g0 - 687;
}//@ assert(x >= y);``` | Add irrelevant Info and rename nodes, dummy relationship between nodes |
| X Original | Increment the original variables by algebraic identities of new variables that equate to 0 and read original variables into the new variables | ```int x = 1, y = 0;
int g0 = 1, g1 = 3;
while (y < 100000) {
x = x + y + (((g0 + g1)*(g0 + g1)) -1*g0*g1 -2*g0*g1)
    - ((g0*g0 + g1*g1) - 1*g0*g1);
y = y + 1 + ((g1*g1 + g0*g0) -1*g1*g0) - (((g1 + g0)*(
    g1 + g0)) -1*g1*g0 -2*g1*g0);
g0 = 178 + x - y;; g1 = g0 - 687 + y - x;
}//@ assert(x >= y);``` | Add irrelevant Info and rename nodes, dummy relationship between nodes |

*Table 7.* Mutations for the `Loop` dataset on an example. For all the programs, $(x = 1 \wedge y = 0) \vee x \geq y \geq 1$ is a valid loop invariant.

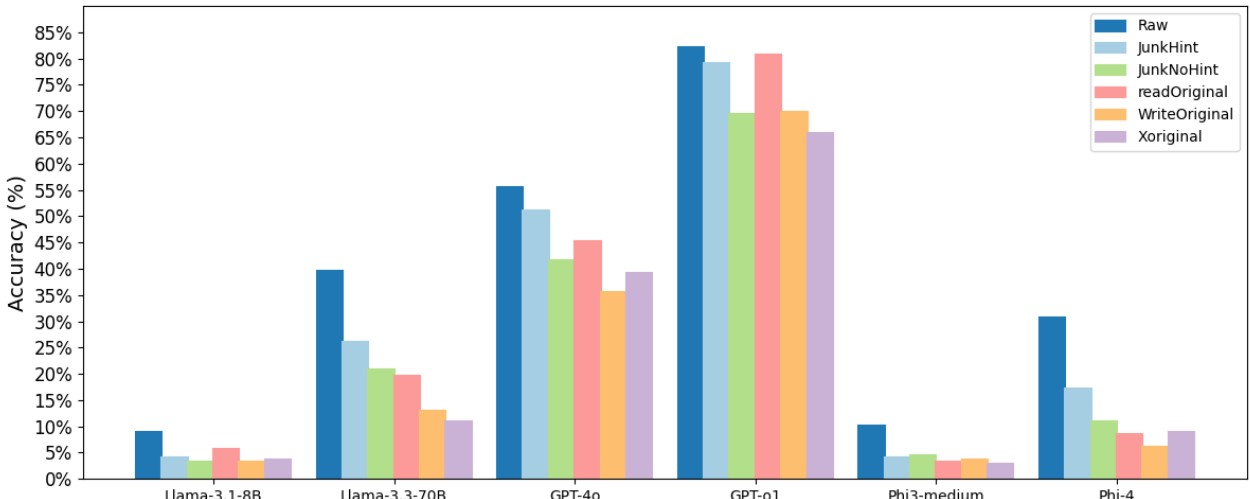

*Figure 25.* Evaluation of various models on the Loop dataset and its mutated versions.

