# OpenReview forum: "RE-IMAGINE: Symbolic Benchmark Synthesis for Reasoning Evaluation"
_ICML.cc/2025/Conference — ICML 2025 poster_

### Official Review · Reviewer_b1Wv · 2025-03-11

**Overall Recommendation:** 3

**Summary:**

This paper introduces R E -I MAGINE: a framework to characterize a hierarchy of reasoning ability in LLMs, alongside an automated pipeline to generate problem variations across all the levels of the hierarchy. By altering problems in an intermediate symbolic representation, RE-IMAGINE generates arbitrarily many problems that are not solvable using memorization alone.  Reductions in performance can be observed when the models are queried with problem variations.

## update after rebuttal
I keep my score positive for this paper.

**Claims And Evidence:**

Not for all.
Problematic claim: these variations have been developed in an ad hoc manner, lacking a systematic hierarchy.
Some datasets also have a systematic hierarchy such as GPQA.

**Essential References Not Discussed:**

None

**Experimental Designs Or Analyses:**

1. the evaluation on the created benchmarks on various domains is somewhat sound.

2. More analyses are needed.
All experiments are about evaluation on the created benchmark (too much content is used for this part). Some analyses should be incorprate to investigate the relationship of question in different level. For example, whether training on level-3 questions can help answering level-2 questions.

More popular LLMs should be adopted for analysis, such as Qwen.

Error studies should be done to analyse why LLMs perform worse on hader questions.

Case studies are also required.

**Methods And Evaluation Criteria:**

It makes sense.

**Other Comments Or Suggestions:**

None

**Other Strengths And Weaknesses:**

None

**Questions For Authors:**

Please refer to the weaknesses above.

**Relation To Broader Scientific Literature:**

It might influence the evaluation of LLMs.

**Theoretical Claims:**

1. the use  of theory on "counterfactual" (for generating level 3 questions)
A counterfactual is a hypothetical scenario that describes what would have happened if a different decision or condition had occurred, contrary to what actually happened in real-world. While in the existing questions, they do not present real-world questions, making the questions of level-3 more like intervention. Maybe creating an scenario which would never happend in real-world could alleviate this issue.

2. The validity of the method might be influenced by the correctness of the NL-symbolic-NL process.
A wrong python program can also produce the same answer as the ground truth.

---

> ### Author Rebuttal · Authors · 2025-04-01
>
> # Response to Claims And Evidence: Some datasets also have a systematic hierarchy such as GPQA
>
> The hierarchy in GPQA represents the difficulty of the **original problems**, whereas the hierarchy we introduce in Re-Imagine defines reasoning complexity through **variations of problems** from existing, well-established benchmarks.
>
> We believe that Level-3 mutations align with counterfactual reasoning based on Pearl’s ladder of causation—**how our belief about the occurrence of event Y changes if event X had value x’ instead of x**. In Figure 1, for example, we present: "Janet bakes muffins with 2 eggs... Assume Janet no longer bakes muffins, how much money does she make every day?" Here, the observed outcome (event Y) is Janet’s daily earnings, while the causal factor (event X) is whether she bakes muffins or not. The logical facts are structured as a math problem, and Level-3 mutations modify one of these facts by introducing an additional assumption.
>
> We agree with the reviewer that creating scenarios unlikely to occur in the real world, such as "Assume Janet bakes muffins using 10,000 eggs every day," can be highly interesting. Thanks to the modular plug-and-play design of our pipeline, we encourage the community to implement mutations of their interest to thoroughly assess models' reasoning abilities.
>
> ---
> # Response to Theoretical Claims 2: NL-symbolic-NL process
>
> We recognize that the accuracy of NL-symbolic-NL translation is critical. Therefore, for benchmarks requiring full NL-symbolic-NL translation, such as GSM8K, we employ four methods to ensure the correctness of the mutated QA pairs and report results with quantified noise:
> - **NL-symbolic:** We select Python solutions that not only produce the correct answer but also ensure that all constant variables in the code (root nodes in the computational graph) align with the numbers in the question, and vice versa (see lines 246-249, left column, in the main paper).
> - **Symbolic-NL:** to ensure the accuracy of the symbolic-to-NL translation, we prompt GPT-4o a second time to back-translate the mutated math problem into Python by modifying the original question’s Python solution. The generated code must produce an execution result that matches the ground truth answer of the mutated question (see line 220-224, right column, in the main paper).
> - **Manual Quality Check:** we manually check the quality of the mutated QA pairs and report the error rate in each mutation (see line 227-235, right column, in the main paper).
> - **Report Results with Quantified Noise:** we show the error rate of the mutated data when reporting the model's performance (see Figure 2).
>
> ---
> # Response to Experimental Designs Or Analyses 2: Error analysis of the relationship of questions across levels, and training on level-3 questions.
>
> We emphasize that this study primarily focuses on introducing the reasoning hierarchy, the benchmark mutation pipeline, and evaluating models' zero-shot and few-shot performance on dataset mutations. Model training is beyond the scope of this paper. However, thanks to the scalability of our proposed pipeline, we are expanding training set mutations to investigate whether exposing models to mutated questions during training can enhance their reasoning abilities. However, the results will not be included in this paper.
>
> In Appendix B.2, we present experiments in an in-context learning setting, revealing that (1) simply replacing the original demonstrations in the context with mutated ones does not improve models’ reasoning accuracy. However, (2) models perform significantly better on generated test set variations when provided with both original and mutated examples as demonstrations. These findings offer insights for future model-training experiments.
>
> ---
> # Response to Experimental Designs Or Analyses 3: More popular LLMs should be adopted for analysis, such as Qwen
>
> We conduct additional experiments with a broader range of popular LLMs, and the observations in the paper still hold for these new models.
>
> **Loop** (mutations Raw, JunkHint, JunkNoHint, readOriginal, WriteOriginal, Xoriginal)
>
> - QwQ-32B:	75.9%	76.7%	66.5%	63.67%	39.59%	36.33%
> - R1-Distill-Llama-70B:	75.10%	62.45%	49.80%	60.41%	49.39%	48.57%
>
> **CruxEval** (mutations Raw, Mutate String (L2), Mutate Value (L2), Redefine Function (L2), Replace Operator (L2), Swap Conditional (L2))
>
> - GPT-4.5:	45.5%	23.55%	32.45%	29.79%	29.14%	32.06%
>
> - GPT-o3-mini:	56.88%	35.95%	59.82%	56.35%	58.52%	50%
>
> **GSM8K** (mutations Raw, SampleValues, OverWriteValue, UselessInfo, AddDependence, InsertConditional)
>
> - qwq-32b 100  98.2  92.6 99.1 59.6 95.6
>
> - r1-distiall-qwen-32b 98.3 93.5  85.9 97.6   63.2  85.0
>
> - GPT-o3-mini 97.4 90.3  84.02   93.5 77.3  91.6
>
> - GPT-4.5 97.45  89.76 81.28 95.47 61.54 89.28
>
> Additional experiments are ongoing. We will include performance results for qwq, r1-distill-qwen, GPT-o3-mini, and GPT-4.5 across all four benchmarks in the camera-ready, and an error analysis in the Appendix.

---

### Official Review · Reviewer_Sq1z · 2025-03-11

**Overall Recommendation:** 3

**Summary:**

To identify whether the performance improvement of LLMs on public benchmarks such as GSM8K indeed comes from the stronger reasoning capabilities or results from mere memorization of training cases, the authors propose RE-IMAGINE to automatically make multi-level modifications to questions in the existing benchmarks. The authors employ RE-IMAGINE to generate problem variations based on 4 benchmarks and observe performance drop of LLMs on the problem variations, which is said to indicate the model reliance on recalling training data.

## Update after rebuttal

Most of my concerns are solved. I have raised my score from 2 to 3.

**Claims And Evidence:**

The paper makes two primary claims: 1) RE-IMAGINE can automatically generate multi-level problem variants, thereby extending existing datasets and mitigating data leakage during training; 2) The performance gap between raw questions and their variants suggests that LLMs rely on recalling training data.

* **Evidence for claim 1:** The authors claim that RE-IMAGINE is a general framework and an automated pipeline. But how to operate the first step, language-to-symbolic transformation, is task-specific and requires manual intervention. And whether or why the mutation mentioned in the paper is general and can be applied to other domain is not clear. The mutation seems to need humans to manually define and restrict the modification types. For example, for the "Sample Values", if the values are numbers, boolean, or strings, the mutation may be different. The authors should provide more details on the generalization of the proposed method and how much manual work is necessary to adopt it in a new dataset. If this framework requires manual redesign of step 1 and step 2 for each new dataset, I think the applicability of this framework is insufficient.

* **Evidence for claim 2:** I doubt whether the experimental results sufficiently support claim 2.

  * Most modifications introduced by RE-IMAGINE significantly increase the complexity of the problems. The UselessInfo operation at level-2 introduces an extra node in the computation graph, and level-3 modifications add reasoning steps. Consequently, the observed decline in model performance could be attributed to either the increased difficulty of the problems or the model's reliance on memorized training data and poor generalization. Previous studies, such as [1] and [2], have shown respectively that irrelevant information and additional reasoning steps can negatively impact model performance.

  * The SampleValues modification at level-2, which involves integer and float values fluctuating within the range of [-10, 10], preserves the problem's difficulty. This suggests that the performance drop in this case may indeed reflect the model's dependence on training data for reasoning. However, this conclusion is not novel, as [3] has already demonstrated in GSM8K that LLMs are sensitive to minor changes in variable values and names, leading to performance degradation.

  * So to robustly support claim 2, the authors need to disentangle the effects of increased problem difficulty from the model's reliance on training data. Without this clarification, the claim remains partially unsubstantiated.

**Essential References Not Discussed:**

The related work section is missing in the paper, and my primary concern is that the unique contribution of this work beyond the following previous work is unclear. While the paper focuses on automatically modifying public datasets, with GSM8K being a central dataset, several highly relevant studies are not discussed or cited. These include:

* GSM-IC [1] shows that irrelevant context can degrade LLM performance.
* iGSM [2] introduces a synthetic pipeline which captures parameter dependencies in a symbolic structure.
* GSM-HARD [9] modify parameter values to be much bigger than in the original dataset.

The contributions beyond the above work and GSM-Symbolic [3] should be more clearly discussed.

**Experimental Designs Or Analyses:**

See Claims and Evidence, ablations for the effects of mutations on question difficulties are needed.

**Methods And Evaluation Criteria:**

RE-IMAGINE supports automated data generation through a pipeline involving: language-to-code transformation, mutations of symbolic representations and code-to-language transformation. The idea is simple and clear. However, the authors should provide more details on the generalization of the proposed method and how much manual work is necessary to adopt it in a new dataset.

The evaluation metric includes model accuracy towards raw questions and the question variations. Besides, in Sec 4.3, the authors also involve the metric of sufficiency / necessity inconsistency, which is proposed in [4]. I recommend that the authors provide a more detailed explanation of the experimental setup and the methodology for this metric in the paper. Currently, the paper merely references [4], which may confuse readers unfamiliar with the prior work.

**Other Comments Or Suggestions:**

Here are some typos:
1. line 019 right: "Traditionally, the **evaluation** of reasoning **evaluation** in LLMs" -> "evaluation of reasoning abilities"
2. line 078 right: "The proposed hierarchy has three levels of increasingly difficulty" -> "increasing difficulty" ?

**Other Strengths And Weaknesses:**

Strengths:
* The paper discusses a crucial problem of how to identify memorization and genuine reasoning in LLMs.
* The presentation of the work is very clear and well-structured, making it accessible to readers.
* Several modifications introduced in RE-IMAGINE, particularly at level-3, appear to be novel to the best of my knowledge. These modifications have the potential to increase the difficulty of the dataset and mitigate the impact of data leakage.

Weaknesses：
* My primary concern revolves around (1) the novelty of the proposed methods.
  * The techniques at level-2 have been previously explored in related works [1, 2, 3].
  * While some methods at level-3 are indeed novel, they do not maintain the same level of problem difficulty as the original tasks. As a result, the observed decline in LLM performance cannot be solely attributed to memorization, as suggested in the 'Claims and Evidence' section. This limits the strength of the conclusions drawn regarding the reliance of LLMs on memorization for answering questions.
* And (2) the generalization of the proposed method. (See the first point in 'Claims and Evidence')

**Questions For Authors:**

1. Please explain about how the pipeline can work accross domains and how much human efforts are needed to adopt this pipeline on different datasets.
2. Please explain more about the cause of performance decreasing. Whether the performance of the model has declined comes from the difficulty of the problem itself, rather than the difference between "memory" and "reasoning".
3. Please explain the novelty of these mutation and your overall framework.
4. How do the three level are related to the three level in causal ladder?

The overall reference:

[1] Large Language Models Can Be Easily Distracted by Irrelevant Context, Shi et al., 2023, https://arxiv.org/abs/2302.00093

[2] Physics of Language Models: Part 2.1, Grade-School Math and the Hidden Reasoning Process, Ye et al., 2024, https://arxiv.org/abs/2407.20311

[3] GSM-Symbolic: Understanding the Limitations of Mathematical Reasoning in Large Language Models, Mirzadeh et al., 2024, https://arxiv.org/pdf/2410.05229

[4] Reasoning Elicitation in Language Models
via Counterfactual Feedback, Hüyük et al., 2024, https://arxiv.org/abs/2410.03767

[5] Counterfactual Memorization in Neural Language Models, Zhang et al., 2023, https://arxiv.org/abs/2112.12938.

[6] Faith and Fate: Limits of Transformers on Compositionality, Dziri et al., 2023, https://arxiv.org/abs/2305.18654

[7] Case-Based or Rule-Based: How Do Transformers Do the Math?, Hu et al., 2024, https://arxiv.org/abs/2402.17709

[8] Towards a Mechanistic Interpretation of Multi-Step Reasoning Capabilities of Language Models, Hou et al., 2023, https://arxiv.org/abs/2310.14491

[9] PAL: Program-aided Language Models, Gao et al., 2023, https://arxiv.org/abs/2211.10435

**Relation To Broader Scientific Literature:**

The paper discusses a crucial problem concerning the evaluation of LLM reasoning capabilities, which is how to identify genuine reasoning and merely recalling training data. This has long been a hotly-debated question in the community of LLM reasoning. Here are some literatures recommended for reference under the topic:
* Reasoning Elicitation in Language Models
via Counterfactual Feedback, Hüyük et al., 2024, https://arxiv.org/abs/2410.03767
* Counterfactual Memorization in Neural Language Models, Zhang et al., 2023, https://arxiv.org/abs/2112.12938.
* Faith and Fate: Limits of Transformers on Compositionality, Dziri et al., 2023, https://arxiv.org/abs/2305.18654
* Case-Based or Rule-Based: How Do Transformers Do the Math?, Hu et al., 2024, https://arxiv.org/abs/2402.17709
* Towards a Mechanistic Interpretation of Multi-Step Reasoning Capabilities of Language Models, Hou et al., 2023, https://arxiv.org/abs/2310.14491

**Theoretical Claims:**

The paper introduces a hierarchical framework to evaluate the reasoning ability of models. While this framework is inspired by the hierarchical structure proposed by Pearl in the context of causality, I find the connection between this framework and causality to be unclear. In other words, could you justify why this particular framework was chosen or how it is fundamentally linked to causality?

---

> ### Author Rebuttal · Authors · 2025-04-01
>
> We thank the reviewer for the detailed feedback.
>
> ---
> # Response to weakness 1: Novelty and the Influence of the Question Difficulty
>
> ## (1) Novelty and contribution:
>
> In summary,
>
> * We present the reasoning ladder for LLMs, which systematically defines different levels of reasoning difficulty. This framework establishes **a unified reasoning hierarchy that integrates both previously studied mutations and the new mutations** introduced in our work. (Please refer to our response to Q1 for further explanation of the reasoning ladder.)
> * Alongside the reasoning hierarchy, we introduce— to the best of our knowledge—**the first scalable mutation generation pipeline that applies across multiple benchmarks and tasks**. This framework enables the creation of an arbitrary number of mutations at each level of the hierarchy for existing benchmark problems. (We elaborate on the scalability further in our response to Weakness 2.)
>
> Compared to the previous studies,
>
> * We pointed out that **previous work is primarily limited to Level-2 mutations** ([1,2,3,9]), which evaluate a model's ability to generalize beyond existing benchmarks while preserving the original reasoning path of the questions.
> * Huyuk et al. [4] explored a level-3 mutation, Bi-Counterfactual. However, like previous studies on level-3 mutations, their approach heavily relies on manually crafted patterns and rule annotations. To the best of our knowledge, **no scalable solution has been proposed for generating problem variations across different reasoning levels spanning multiple benchmarks and tasks**.
>
> ## (2) Disentangle the decline in model performance from the question difficulty:
>
> We use GSM8K as an example. We quantitatively define the difficulty of a numerical question reasoning using the number of calculation steps in the code snippet, following iGSM [2]. We compute the average accuracy of each model across examples with varying numbers of calculation steps. Given the large number of models tested, we aggregate their results and present the overall average accuracy in the following table. We included detailed results for each model respectively in the Appendix of the paper.
>
> | Intervention Type | 2 steps | 3 steps | 4 steps | 5 steps | 6 steps |
> |-------------------|------|------|------|------|------|
> | Raw | 0.95 | 0.94 | 0.84 | 0.91 | 0.83 |
> | SampleValues | 0.87 | 0.84 | 0.75 | 0.74 | 0.80 |
> | UselessInfo | 0.91 | 0.90 | 0.90 | 0.81 | 0.88 |
> | CounterFactual | 0.74 | 0.71 | 0.75 | 0.62 | 0.67 |
> | InsertConditional | 0.62 | 0.68 | 0.65 | 0.61 | 0.57 |
> | AddDependence | 0.57 | 0.47 | 0.46 | 0.45 | 0.42 |
>
> From the table we make three key observations:
>
> * In nearly all scenarios, **even when tested on examples with the same number of calculation steps, models consistently perform worse on the mutated sets compared to the original test set**.
> * Compared to Level-2, Level-3 mutations present a significantly greater challenge. Especially, **the accuracy on Level-3 mutations with just two calculation steps is lower than on Raw test examples with six calculation steps with a significant margin.**
>
> ---
> # Response to weakness 2: Generalization of the method
>
> The pipeline requires three types of adapters: Question-to-Symbolic Adapters, Symbolic Representation-to-Mutation Adapters, and Mutation-to-Natural Language Question Adapters. The pipeline is designed in a **modular plug-and-play fashion**, making **all adapters both reusable and customizable**. Users can either utilize existing adapters if they meet their needs or modify them to suit their specific dataset or domain. The estimated manual effort required for **adapter customization** is based on the four domains we implemented:
>
> * Question to Symbolic Adaptor: **Prompts Writing**.
> * Symbolic Representation to Mutation Adapter: **Write around 50 lines of code** to define the mutation of the symbolic representation.
> * Mutation to Natual Language Question Adaptor: **Write around 100 lines of code for model prompting** to translate the mutation to natural language.
>
> **We replaced the word "automatic" with the more precise term "scalable" in the paper.**
>
> ---
> # How do the three levels relate to the causal ladder?
>
> The key connection between Pearl's ladder and our framework is the problem's computation graph, which can be understood as a causal model in Pearl's framework. Each problem in a benchmark can be interpreted as a single realization of the graph with specific node values. Experiment associated with different perturbations in such graph can be related to operations in Pearl's ladder of causation. For instance, computing the effect in the outcome of the change in one leave node maps to the definition of a counterfactual. Note, however, that that not all mutations in the three levels have a causal counterpart (like adding an irrelevant piece of information or changing an operation). In this sense our framework can cover a broader definition of reasoning in each level.

---

> > ### Comment · Reviewer_Sq1z · 2025-04-03
> >
> > I appreciate your additional experiments and justifications, especially about `Disentangle the decline in model performance from the question difficulty`. I want to further confirm how you define "steps" in this experiment. I am still concerned that the number of calculation steps do not actually represent the genuine difficulty. For example, in the case of “UselessInfo”, adding an irrelevant node does not increase the number of calculation steps, but it does make the question harder as shown in [1].
> >
> > My understanding about the definition of the calculation step is as follows (I use Figure 1 in your original paper as an example):
> > * SampleValues: the difficulty is maintained, no problem.
> > * UselessInfo: add extra node, but not increase calculation step?
> > * AddDependence: add 1 extra node, increase 1 calculation step, no problem.
> > * InsertConditional: add 1 extra node, I’m not sure how the number of calculation steps change.
> > * CounterFactual: add 1 extra node, increase 1 calculation step, no problem.
> > * Bi-CounterFactual: add 3 extra node, I’m not sure how the number of calculation steps change.

---

> > > ### Author Response · Authors · 2025-04-03
> > >
> > > We thank the reviewer's response!
> > >
> > > The reviewer's interpretation of the calculation step was mostly accurate. We provide further clarification on the three mutations that the reviewer is uncertain about:
> > >
> > > * **UselessInfo**
> > >
> > >   **[Mutated question]**: Janet’s ducks lay 10 eggs per day. She eats 4 for breakfast and bakes muffins with 2. She sells the remainder for $3 per fresh duck egg. **Janet plans to save these money for a new dress.** How much in dollars she makes every day?
> > >
> > >   **[Mutated code]**:
> > >   ```python
> > >   eggs = 10
> > >   breakfast_eggs = 4
> > >   muffin_eggs = 2
> > >   remainder = eggs - breakfast...
> > >   price = 3
> > >   sales = price * remainder
> > >
> > >   # The useless Info
> > >   # Note that, the added number is sampled. In other examples, it may not be 0.
> > >   # In Figure-1 in the paper, we missed '+0'. We already updated the figure.
> > >   dress = sales + 0
> > >
> > >   return sales
> > >   ```
> > >   **[Difficulty Changes]**: The mutation adds an extra calculation step. To clarify, in the table, all examples in UselessInfo contain **one additional reasoning step** compared to the Raw examples. However, this extra step does not impact the calculation of the final answer. Therefore, according to our definition, we still consider UselessInfo to be a **Level-2 mutation**.
> > >
> > >
> > > * **InsertConditional**
> > >
> > >   **[Mutated question]**: Janet’s ducks lay 10 eggs per day. She eats 4 for breakfast and bakes muffins with 2. She sells the remainder for $3 per fresh duck egg. **Janet only sells eggs if her ducks lay at least 16 eggs in a day.** How much in dollars she makes every day?
> > >
> > >   **[Mutated code]**:
> > >   ```python
> > >   eggs = 10
> > >   breakfast_eggs = 4
> > >   muffin_eggs = 2
> > >   remainder = eggs - breakfast...
> > >   price = 3
> > >
> > >   # The inserted Condition
> > >   if eggs > 16:
> > >       sales = price * remainder
> > >   else:
> > >       salse = 0
> > >
> > >   return sales
> > >   ```
> > >   **[Difficulty Changes]**: Compared to the original problem, the mutation introduced an if-else operation. To clarify, in the table, all examples in InsertConditional contain **one additional reasoning step** compared to the Raw examples.
> > >
> > > * **Bi-CounterFactual** is equivalent to **CounterFactual**, as in our PN/PS experiments, the 'Raw' questions were also transformed into binary questions. Therefore, the additional reasoning step introduced by Bi-CounterFactual is the assumption of a value change.
> > >
> > > We want to emphasize that testing reasoning is inherently an adversarial task. A model capable of reasoning through a problem should perform equally well across a diverse range of its variations. Our work paves the way for defining and implementing these variations at scale. In our paper, **we presented multiple variations designed to evaluate the model’s reasoning ability from different perspectives, going beyond just differentiating it from memorization** (SampleValues). For example:
> > > * UselessInfo: assesses the model’s ability to disregard irrelevant information.
> > > * Level-3 mutations: assess the models’ ability to accurately integrate new information and logic into existing problems. While these mutations introduce an additional operation that could increase difficulty, we argue that:
> > >   * (1) if a model has truly learned basic math, adding one more step should not significantly alter the problem’s complexity, and
> > >   * (2) the ability to envision an alternative scenario is a fundamental aspect of reasoning, which these mutations are specifically designed to test.
> > >
> > > We promise to include the discussion about the problem difficulties in the updated version of the paper.

---

### Official Review · Reviewer_RCQD · 2025-03-11

**Overall Recommendation:** 3

**Summary:**

The paper mainly introduces a benchmark synthesis pipeline for math and coding reasoning problem. The proposed pipeline can make modifications of the original benchmark (question, answer) pairs to make it a different (potentially more challenging) instance. The main motivation is to evaluate the true reasoning ability of the models other than memorizing training set. The pipeline comprises 3 steps: (1) NL-to-code transformation (2) make changes to the code/computation graph (3) transform back from code to NL. The results show that models usually yield worse performance on the synthesis benchmarks.

## Update after rebuttal
Overall I think this work is a moderate extension of the Gsm-symbolic paper; I will keep my already possitive rating

**Claims And Evidence:**

The claim is: fixed benchmarks are not good enough as there can be leakage, so we need this benchmark systhesis tool to alter the original benchmark to get more faithful evaluation results

I feel the "sysnthesis" part is clear and the results well supports the claim, we observe quite some drops when altering the original question, even if it is just changing values or adding irrevant information; (earlier work such as [1] also have this observation and applied similar techniques);

However, the claim that this can be a useful "benchmark" is unclear; the main goal of a benchmark is to evaluate performance accross different models, however, if the synthesis is not deterministic, it is not possible to compare results with previously reported scores from other models; if we fix the synthesis, then the problem goes back to a "fixed" benchmark with potential leakage once released;
If the authors can show that even though the synthesis is not deterministic, some metric e.g., the drop of performance, can still be robust enough to have comparison across different models, the contribution of this paper would be more clear.

---
Reference:
[1] Mirzadeh, Iman, et al. "Gsm-symbolic: Understanding the limitations of mathematical reasoning in large language models." arXiv preprint arXiv:2410.05229 (2024).

**Essential References Not Discussed:**

N/A

**Experimental Designs Or Analyses:**

Solid

**Methods And Evaluation Criteria:**

Some important technical details seems to be not explained (or hard to find), this includes: What exact models are used for

(1) NL-to-code

(2) Computation graph parsing

(3) code-to-NL

**Other Comments Or Suggestions:**

N/A

**Other Strengths And Weaknesses:**

N/A

**Questions For Authors:**

N/A

**Relation To Broader Scientific Literature:**

This is related to, and can be viewed as an expansion of: Mirzadeh, Iman, et al. "Gsm-symbolic: Understanding the limitations of mathematical reasoning in large language models." arXiv preprint arXiv:2410.05229 (2024).

**Theoretical Claims:**

N/A

---

> ### Author Rebuttal · Authors · 2025-04-01
>
> # Response to Claims And Evidence: Robustness to stochastic synthesis
>
> We believe there may be a misunderstanding about the experimental setup. To clarify:
>
> * **All models in the paper are tested on the same generated benchmark instantiation**, ensuring fair comparison between models.
> * We also report the **statistical accuracy of models on GSM8K numerical answer predictions in Figure 7**. The bar plot presents the average accuracy and variance for each model, tested on 10 test variations per mutation, sampled using 10 different seeds. Importantly, these 10 test variations are identical across all models in this experiment. The results show the robustness of the metric to stochastic synthesis.
>
> We thank the reviewer for pointing out the unclear instructions on how to evaluate using dynamic benchmarks like Re-Imagine. We will add the following clarification to the paper:
>
> *For a fair comparison, all models should be tested using the same data. In practice, to evaluate using dynamic benchmarks like Re-Imagine, we recommend that researchers: (1) sample multiple test variations initially; (2) record and report the random seeds in publications or repositories; (3) report the statistical accuracy on the sampled test variations for both baseline models and proposed approaches.*
>
> # Response to Methods And Evaluation Criteria: What exact models are used for (1) NL-to-code (2) Computation graph parsing (3) code-to-NL?
>
> We use language models only in the NL-to-code and code-to-NL steps. All mutations to symbolic representations (ie. computation graphs) are performed explicitly by modifying the AST (abstract syntax tree https://docs.python.org/3/library/ast.html) data structure of the code according to user-defineable rules.
>
> For NL-to-code and code-to-NL:
>
> * GSM8K
>   * NL-to-code uses Mixtral-8x7B (see line 240-244, left-column in the main paper).
>   * Code-to-NL uses GPT-4o (see line 266-270, left-column in the main paper, Figure 13 and 14 in Appendix B).
>
> * CLadder
>   * NL-to-code uses the causal engine offered in the original CLadder benchmark (see line 1124 in Appendix C).
>   * Code-to-NL uses Meta-Llama70BInstruct (see line 1140 in Appendix C).
>
> Loop and CruxEval start from the symbolic representation, so the NL-to-code and Code-to-NL steps are skipped; thus, applying Re-Imagine to these benchmarks does not require use of any models.

---

> > ### Comment · Reviewer_RCQD · 2025-04-03
> >
> > (copying official comment here so authors can see this) Thanks authors for the rebuttal. Overall I think this work is a moderate extension of the Gsm-symbolic paper; I will keep my already possitive rating

---

### Official Review · Reviewer_2n8z · 2025-03-14

**Overall Recommendation:** 4

**Summary:**

This work creates a framework for LLM reasoning evaluationexpands and scales up LLMs reasoning evaluation by means of an automated pipeline that converts benchmark problems into symbolic representations and then back again, and a 'mutations' step to create variations in pre-existing questions to further test reasoning capabilities.

**Claims And Evidence:**

The mutation step in particular offers a great way to augment data and benchmarking for LLMs. This is a versatile tool that uses powerful symbolic reasoning tactics to test reasoning.

**Essential References Not Discussed:**

I think one aspect you didn't really cover is the adversarial nature of a lot of the testing that is done in this field. The mutations could include something like robustness testing. Also, similar works to be cited would be DreamCoder and SATNet.

**Experimental Designs Or Analyses:**

The work does not do experimentation on how models perform on their framework versus other reasoning evaluation methods (like those referenced in the related works section from from Mirzadeh et al., Lewis & Mitchell, or Gonzalez & Nori). Extremely thorough model testing (Figure 2) and benchmark usage (code benchmarks).

**Methods And Evaluation Criteria:**

Because the mutation part to me is the best methodological novelty of this paper, I would personally like to see more comprehensive testing on the mutations, apart from the selective manual verification you did.

**Other Comments Or Suggestions:**

I think pages 2-4 need some editing. There seems to be accidentally duplicated paragraphs (the Judea Pearl quotes, the definitions of the layers etc)

**Other Strengths And Weaknesses:**

Weaknesses: should directly compare against related works methods to show concrete improvements.
Strength: very thorough testing of diverse domains, datasets, ablations (in appendix) and mutation types. Interesting findings about complex reasoning and weaknesses of these LLMs

**Questions For Authors:**

All prompts are done with zero-shot prompting. Did you look at all at few-shot?

**Relation To Broader Scientific Literature:**

Generally speaking, the thing I find least convincing is the reason *why* this framework is needed overall. The mutation step is interesting, but is the symbolic conversion approach better than simpler mutation methods? This isn't particularly addressed experimentally. Of course, it makes sense logically that symbolic conversion would allow for better code/math mutations. But this can be formally shown.

**Theoretical Claims:**

None

---

> ### Author Rebuttal · Authors · 2025-04-01
>
> # Response to Experimental Designs Or Analyses and Weaknesses: should directly compare against related works methods to show concrete improvements
>
> We highlight that **the goal of Re-Imagine is to establish a unified reasoning hierarchy that integrates both previously studied mutations and the new mutations introduced in our work**.
>
> We re-implement the two most widely studied mutations from previous work—UselessInfo and SampleValues—while scaling them up to the entire GSM8K test set. Since no prior studies have focused on evaluating models' reasoning abilities through mutations in the other three benchmarks—Loop, CruxEval, and CLadder—there is no meaningful mutations to be included in the framework.
>
> ---
> # Response to Experimental Relation To Broader Scientific Literature: Why Symbolic Representation?
>
> The primary reason for introducing the executable symbolic representation is to obtain ground truth answers for the mutated examples in a deterministic manner. For instance, if the symbolic representation is a Python code snippet, the answer to the mutated question is derived by executing the mutated code.  As a result, both the symbolic-to-mutation and mutation-to-answer modules in the pipeline are deterministic. The other two modules—NL-to-symbolic and mutation-to-NL question—rely on LLMs. Given that LLMs have demonstrated relatively high reliability and robustness in NL-to-code and code-to-description tasks, the accuracy of these two modules is generally satisfactory. On top of these, for each LLMs-based module, we conduct verifications to further guarantee the mutations accuracy:
> * NL-to-symbolic: see lines 246-249, left column, in the main paper.
> * Symbolic-to-NL question: see line 220-224, right column, in the main paper.
>
> **In conclusion, executable symbolic representations enable us to break down the mutation process into smaller steps that are either deterministic or easily handled by LLMs, and can be thoroughly verified.** In addition, the symbolic representation gives us greater control over the mutation generation process. **We can explicitly define and study different types of mutations.**
>
> We are unclear about which "simpler mutation methods" the reviewer is referring to. We assume they might be methods that generate mutated QA pairs directly using LLMs without the assistance of symbolic representations. However, as shown in the paper, LLMs struggle with answering mutated questions, and relying on them to generate ground truth answers for mutated questions can be risky. Additionally, there are no reliable validation methods to verify the accuracy of the generated mutations. We are open to further discussion if the reviewer provides more details about the simpler mutation methods they have in mind.
>
> ---
> # Response to Experimental Questions For Authors: Zero-shot Only?
>
> For all four benchmarks, we adopt the standard in-context learning methods commonly used in the original benchmarks. In previous studies, zero-shot learning has been most widely used in CLadder, CruxEval, and Loop, while 8-shot learning has been the most common setup for GSM8K. We replicate this configuration in our experiments. In Appendix B.2, we further extend our experiments to explore the impact of different types of in-context learning examples on answering mutated GSM8K questions.
>
> ---
> # Response to repeated text in pages 2-4
>
> We did not find any repeated paragraphs or quotations as mentioned by the reviewer (any specific line number references to this would be greatly appreciated); however, we appreciate the note and have performed an additional pass editing this section for proofreading and clarity. We acknowledge that the typesetting of the table and caption may have led to some confusion or re-reading of earlier text, and we improve the layout for the camera-ready submission.

---

### Decision · Program_Chairs · 2025-05-01

**Decision:**

Accept (poster)

**Comment:**

The paper gives a framework to synthesize benchmarks for coding and math reasoning tasks. The framework involves: (i) converting natural language problem statements to a code representation, (ii) applying mutations to this representation, and (iii) transforming the problems back from code to natural language. The approach is used to evaluate several popular LLMs. It is shown that models usually perform worse on the mutated benchmark. The conclusion is that LLMs often recall on statistical recall when performing "reasoning" tasks.

The reviewers generally agree that the approach is reasonable and the claims are sound. (A comparison with direct mutations to natural language problem statements would have made the paper stronger but this is not a major weakness.) The approach isn't particularly surprising, and previous work has studied ideas closely related to the paper's. However, the paper deserves credit for connecting these ideas into a unified framework, and the connection to Pearl's causal ladder is also appealing. I would put this paper in the "accept if there's room" category.